**ARTICLES**
## OPEN
# Genetic correlates of phenotypic heterogeneity in autism

Varun Warrier [1] ✉, Xinhe Zhang[1], Patrick Reed[1], Alexandra Havdahl [2,3,4], Tyler M. Moore [5,6], Freddy Cliquet[7], Claire S. Leblond[7], Thomas Rolland[7], Anders Rosengren [8,9], EU-AIMS LEAP*, iPSYCH-Autism Working Group, Spectrum 10K and APEX Consortia, David H. Rowitch[10], Matthew E. Hurles [11], Daniel H. Geschwind [12,13,14,15], Anders D. Børglum [8,16,17], Elise B. Robinson[18,19,20], Jakob Grove [8,16,17,21], Hilary C. Martin [11], Thomas Bourgeron [7,40] and Simon Baron-Cohen [1,40] ✉

The substantial phenotypic heterogeneity in autism limits our understanding of its genetic etiology. To address this gap, here we investigated genetic differences between autistic individuals ($n_{max} = 12,893$) based on core and associated features of autism, co-occurring developmental disabilities and sex. We conducted a comprehensive factor analysis of core autism features in autistic individuals and identified six factors. Common genetic variants were associated with the core factors, but de novo variants were not. We found that higher autism polygenic scores (PGS) were associated with lower likelihood of co-occurring developmental disabilities in autistic individuals. Furthermore, in autistic individuals without co-occurring intellectual disability (ID), autism PGS are overinherited by autistic females compared to males. Finally, we observed higher SNP heritability for autistic males and for autistic individuals without ID. Deeper phenotypic characterization will be critical in determining how the complex underlying genetics shape cognition, behavior and co-occurring conditions in autism.

The core diagnostic criteria for autism consist of social communication difficulties, unusually restricted and repetitive behavior, and sensory difficulties that are present early in life and affect social, occupational and other important domains of functioning[1,2]. However, these criteria are broad, leading to substantial heterogeneity. Two individuals with very different phenotypic features, co-occurring conditions, support needs or outcomes may both be diagnosed as autistic[1,3].

Heterogeneity in autism can arise from multiple, partly overlapping sources. This includes differences in core diagnostic features (core features)[1,3,4] and associated features such as IQ, adaptive behavior and motor coordination, all of which have an impact on life outcomes[3,5,6]. Furthermore, sex and gender[7,8] and co-occurring ID and developmental, behavioral and medical conditions[9,10] alter the presentation and measurement of core autism features. While a few studies have attempted to investigate the genetic influences on this heterogeneity[11–18], substantial gaps remain. First, existing studies investigating genotype–phenotype associations have been limited to summed scores of core autism features in smaller sample sizes[19–21] rather than the underlying latent dimensions. This distinction is important given that autism is phenotypically dissociable[12,22,23], and some associations may emerge only when latent traits are considered. Second, while the impact of de novo genetic variants on co-occurring developmental disabilities is reasonably well characterized[17,20,21], the impact of common genetic variants is unknown. Third, although sex differences in autism vary by the presence of ID[17,24,25], the sex-differential impact of common genetic variants in autistic individuals with and without ID is unknown. Finally, the impact of latent core autism phenotypes, sex and de novo variants on the common variant heritability also warrants investigation with large sample sizes.

Here, we address these four questions by combining genetic and phenotypic data from up to 12,893 autistic individuals from four different datasets. We focus on de novo protein-truncating and missense variants in constrained genes (high-impact de novo variants)[17,26] and PGS for autism and genetically correlated phenotypes[16].

[1]Autism Research Centre, Department of Psychiatry, University of Cambridge, Cambridge, UK. [2]Nic Waals Institute, Lovisenberg Diaconal Hospital, Oslo, Norway. [3]Department of Mental Disorders, Norwegian Institute of Public Health, Oslo, Norway. [4]PROMENTA Research Center, Department of Psychology, University of Oslo, Oslo, Norway. [5]Department of Psychiatry, University of Pennsylvania, Philadelphia, PA, USA. [6]Lifespan Brain Institute of the Children's Hospital of Philadelphia and University of Pennsylvania, Philadelphia, PA, USA. [7]Human Genetics and Cognitive Functions, Institut Pasteur, UMR3571 CNRS, Université de Paris Cité, Paris, France. [8]The Lundbeck Foundation Initiative for Integrative Psychiatric Research, iPSYCH, Aarhus, Denmark. [9]Institute of Biological Psychiatry, MHC Sct Hans, Copenhagen University Hospital, Copenhagen, Denmark. [10]Department of Paediatrics, Cambridge University Clinical School, Cambridge, UK. [11]Human Genetics Programme, Wellcome Sanger Institute, Wellcome Genome Campus, Hinxton, UK. [12]Program in Neurobehavioral Genetics, Semel Institute, David Geffen School of Medicine, University of California, Los Angeles, Los Angeles, CA, USA. [13]Department of Neurology, Center for Autism Research and Treatment, Semel Institute, David Geffen School of Medicine, University of California, Los Angeles, Los Angeles, CA, USA. [14]Department of Psychiatry, Semel Institute, David Geffen School of Medicine, University of California, Los Angeles, Los Angeles, CA, USA. [15]Department of Human Genetics, David Geffen School of Medicine, University of California, Los Angeles, Los Angeles, CA, USA. [16]Center for Genomics and Personalized Medicine (CGPM), Aarhus University, Aarhus, Denmark. [17]Department of Biomedicine (Human Genetics) and iSEQ Center, Aarhus University, Aarhus, Denmark. [18]Stanley Center for Psychiatric Research, Broad Institute of MIT and Harvard, Cambridge, MA, USA. [19]Analytic and Translational Genetics Unit, Department of Medicine, Massachusetts General Hospital, Boston, MA, USA. [20]Department of Epidemiology, Harvard T.H. Chan School of Public Health, Boston, MA, USA. [21]Bioinformatics Research Centre, Aarhus University, Aarhus, Denmark. *Lists of authors and their affiliations appear at the end of the paper. ✉e-mail: vw260@medschl.cam.ac.uk; sb205@cam.ac.uk

1293

Finally, this larger sample size alongside more detailed information on genes underlying severe developmental disorders[27] also allows us to revisit and provide deeper insights into two additional important issues relevant to heterogeneity in autism: the association of high-impact de novo variants with (1) co-occurring developmental disabilities and (2) sex.

## Results

**Identifying latent phenotypes in core autism features.** A critical challenge in identifying sources of heterogeneity in autism is understanding the latent structure of core autism phenotypes. To this end, we combined two widely used parent-reported measures of autistic traits (Repetitive Behavior Scale—Revised (RBS)[28] and Social Communication Questionnaire—Lifetime version (SCQ)[29]) for 24,420 autistic individuals from the Simons Simplex Collection (SSC)[30] and the Simons Foundation Powering Autism Research for Knowledge (SPARK)[31] cohorts.

In exploratory factor analyses (Methods), we tested 42 different factor models, including bifactor models (Supplementary Table 1 and Supplementary Fig. 1). We identified a correlated six-factor model with good theoretical interpretation (Supplementary Fig. 2), and confirmatory factor analyses identified fair fit indices (confirmatory fit indices, 0.92–0.94; Tucker–Lewis indices (TLI), 0.92–0.94; root mean square errors, 0.056–0.060). Fit indices increased modestly when including orthogonal method factors in the model (Supplementary Table 1). The explained common variances and hierarchical $\Omega$ values for the bifactor models were low (<0.8), suggesting that general factors may not explain the data well (Supplementary Table 2). The six identified factors are (1) insistence on sameness (F1), (2) social interaction at the age of five years (F2), (3) sensory–motor behavior (F3), (4) self-injurious behavior (F4), (5) idiosyncratic repetitive speech and behavior (F5) and (6) communication skills (F6) (Supplementary Table 3). These broadly correspond to four restricted, repetitive and sensory behavior factors, that is, non-social factors (insistence of sameness, sensory–motor behavior, self-injurious behavior and idiosyncratic repetitive speech and behavior) and two social factors (social interaction and communication skills).

All interfactor correlations were significant and moderate to high in magnitude, with higher correlation among non-social and social factors than between social and non-social factors (Fig. 1a). Sex differences were minimal (Cohen's $d < 0.1$; Fig. 1b and Supplementary Table 4a). All factors were negatively correlated with full-scale IQ (Fig. 1c, Supplementary Fig. 3 and Supplementary Table 4b). In this cross-sectional data, older participants had lower factor scores (that is, fewer difficulties), with the exception of 'social interaction' (Fig. 1d), in line with previous research[32]. Alternatively, this could reflect diagnostic bias. However, of the 21 items in the 'social interaction' factor, 19 specifically ask about behavior between the ages 4 and 5 years (Methods), and this trajectory likely reflects recall bias, as caregivers are likely to report more severe behaviors retrospectively[33]. Similar trends were observed in both males and females (Supplementary Fig. 4). Of the six factors and RBS and SCQ, only insistence on sameness (F1) and self-injurious behavior (F4) had significant SNP heritability (Supplementary Table 5). There were moderate to high genetic correlations among the six factors (Supplementary Table 6).

**Common genetic variants are associated with core autism features.** We next conducted association analyses between 19 different core and associated features and different classes of genetic variants (Methods). We first investigated the association between the 19 features and PGS for autism (iPSYCH autism data freeze), intelligence[34], educational attainment[35], attention-deficit–hyperactivity disorder (ADHD)[36] and schizophrenia[37] and, as a negative control, hair color[38] ($n = 2,421$–12,893, Supplementary Table 7). In multiple

regression analyses, ADHD PGS were associated with increased non-social core autism features (total scores on the RBS, insistence on sameness, sensory–motor behavior and self-injurious factor scores) (Fig. 2 and Supplementary Table 8). Intelligence PGS were associated with increased full-scale and nonverbal IQ. Educational attainment PGS were associated with increased full-scale and verbal IQ and reduced scores on core autism features. Schizophrenia PGS were associated with reduced adaptive behavior, measured using the composite score of the Vineland Adaptive Behavior Scales. Moderate heterogeneity ($I^2 > 50\%$) was observed only for 10% of the associations. The majority of the significant associations (12 of 15) had concordant effect directions in all cohorts (Supplementary Fig. 5). We did not identify any significant genotype–phenotype association using hair color (blonde versus other) as a negative control (Supplementary Table 8).

In line with previous results[17,20,21], the number of high-impact de novo variants (protein-truncating single-nucleotide variants (SNVs) and structural variants and missense variants with missense badness, PolyPhen-2 and constraint (MPC) score >2, $n = 2,863$–4,442) was associated with reduced measures of IQ, adaptive behavior and motor coordination but not core autism features (Fig. 2 and Supplementary Table 9). The effect sizes of the PGS were not attenuated after controlling for the presence of high-impact de novo variants (Supplementary Table 9), which was true even for full-scale IQ.

In autistic individuals, full-scale IQ decreased with increasing number of high-impact de novo variants but increased with increasing PGS for intelligence (Fig. 3a). No strong evidence of interaction between PGS for intelligence and high-impact de novo variants was observed, suggesting their additive effects on full-scale IQ. Among the significant genotype–phenotype associations, accounting for full-scale IQ did not attenuate the effects of PGS on core autism features (Fig. 3b and Supplementary Table 10), which was supported by minimal and statistically non-significant genetic correlations between full-scale IQ and the core autism features (Supplementary Table 6). By contrast, associations between high-impact de novo variants and associated autism features were attenuated, partly because of the moderate phenotypic correlations between these features and full-scale IQ (Fig. 4c).

**Core autism phenotypes in high-impact de novo carriers.** While high-impact variants in some autism-associated genes lead to core autistic features, notably in animal models (for example, refs. [39,40]), as a group, they were not robustly associated with core autism features in this study (Fig. 2). It is unclear whether the latent structure of core phenotypes differs in autistic individuals with high-impact de novo variants (henceforth, carriers) compared to autistic individuals without any known high-impact de novo variant (henceforth, non-carriers). We thus investigated differences in the latent structure of core autism phenotypes between carriers ($n = 325$) and non-carriers ($n = 2,727$). Although likelihood-ratio tests identified significant configural invariance violation (that is, the factor structure dissimilar across groups, $P < 2 \times 10^{-16}$), this was due to the relatively large sample size: the fit indices and visual inspections of the latent structure suggested that the differences were minimal (Supplementary Table 11).

Given this, we first investigated whether autistic carriers had higher PGS for autism than non-carriers, which may account for core autism features in carriers (additivity). As demonstrated previously but with a different set of PGS[19], autistic carriers had lower PGS for autism than autistic non-carriers ($\beta_{PGS} = -0.16$, s.e. = 0.045, $P = 3.67 \times 10^{-4}$, linear regression; Fig. 4a). This difference was not observed for PGS for educational attainment, IQ or schizophrenia (Supplementary Table 12). However, while autistic non-carriers had higher PGS than non-autistic siblings ($\beta_{PGS} = 0.19$, s.e. = 0.023, $P = 2.68 \times 10^{-15}$, logistic regression), autistic carriers ($n = 579$) were indistinguishable from non-autistic siblings ($n = 3,681$) based on

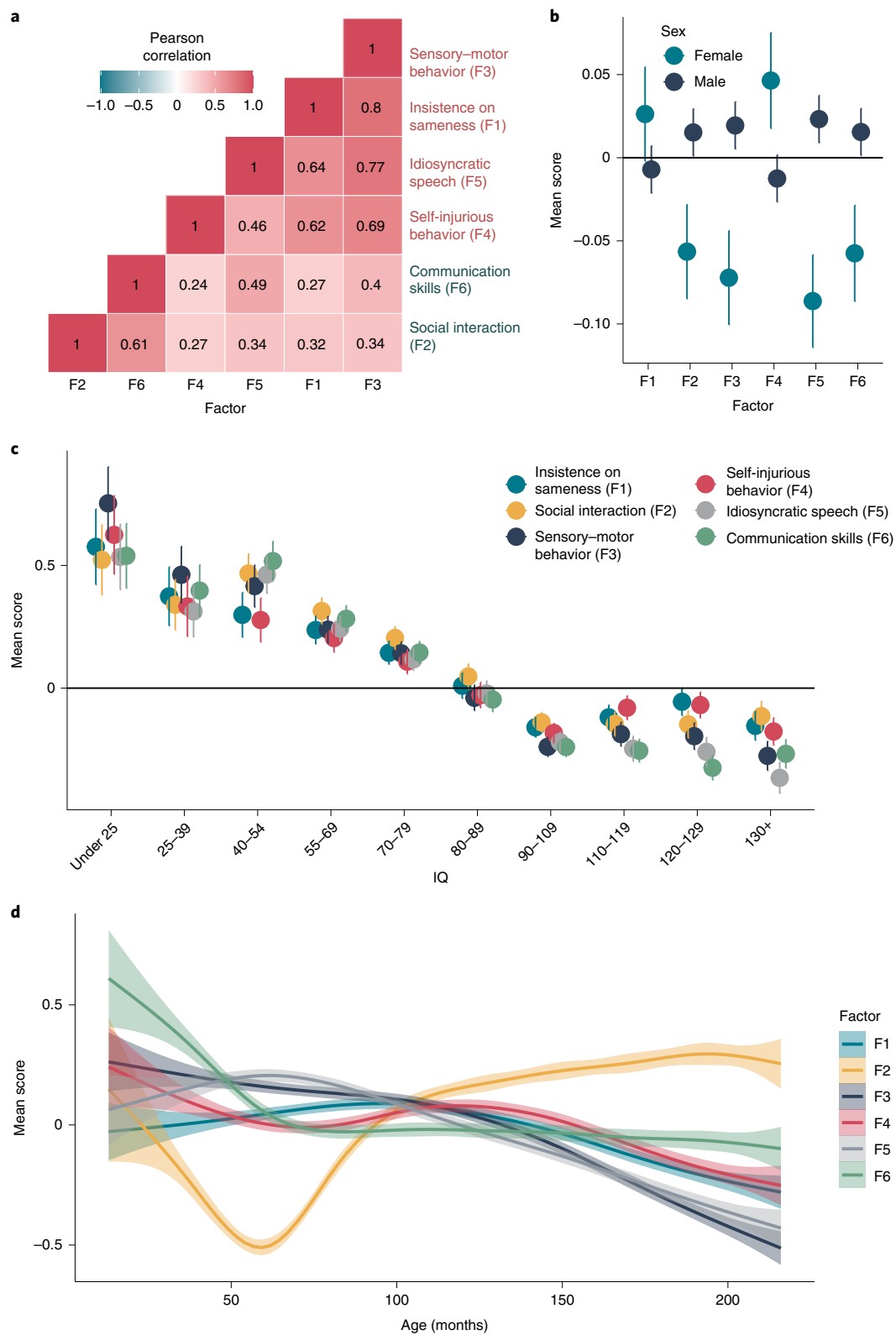

**Fig. 1 | Factor analyses of the core autism features. a**, Pearson's correlation coefficient between the six factors. Factors are ordered based on hierarchical clustering, demonstrating higher correlations among social (teal labels) and non-social (red labels) factors than between them. **b**, Mean scores and 95% confidence intervals for the six factor scores in males ($n = 18,761$) and females ($n = 5,050$). **c**, Mean scores and 95% confidence intervals for the six factor scores in ten full-scale IQ bins ($n = 11,371$). **d**, Smoothed Loess curve for mean factor scores for the six factors across age. The six factors are (1) insistence of sameness (F1), (2) social interaction (F2), (3) sensory–motor behavior (F3), (4) self-injurious behavior (F4), (5) idiosyncratic repetitive speech and behavior (F5) and (6) communication skills (F6). F2 primarily consists of items related to social interaction at the age of 4–5 years (past 12 months if younger than 4 years); hence, the trajectory likely reflects recall bias in participants. Shaded regions indicate 95% confidence intervals of the Loess curves.

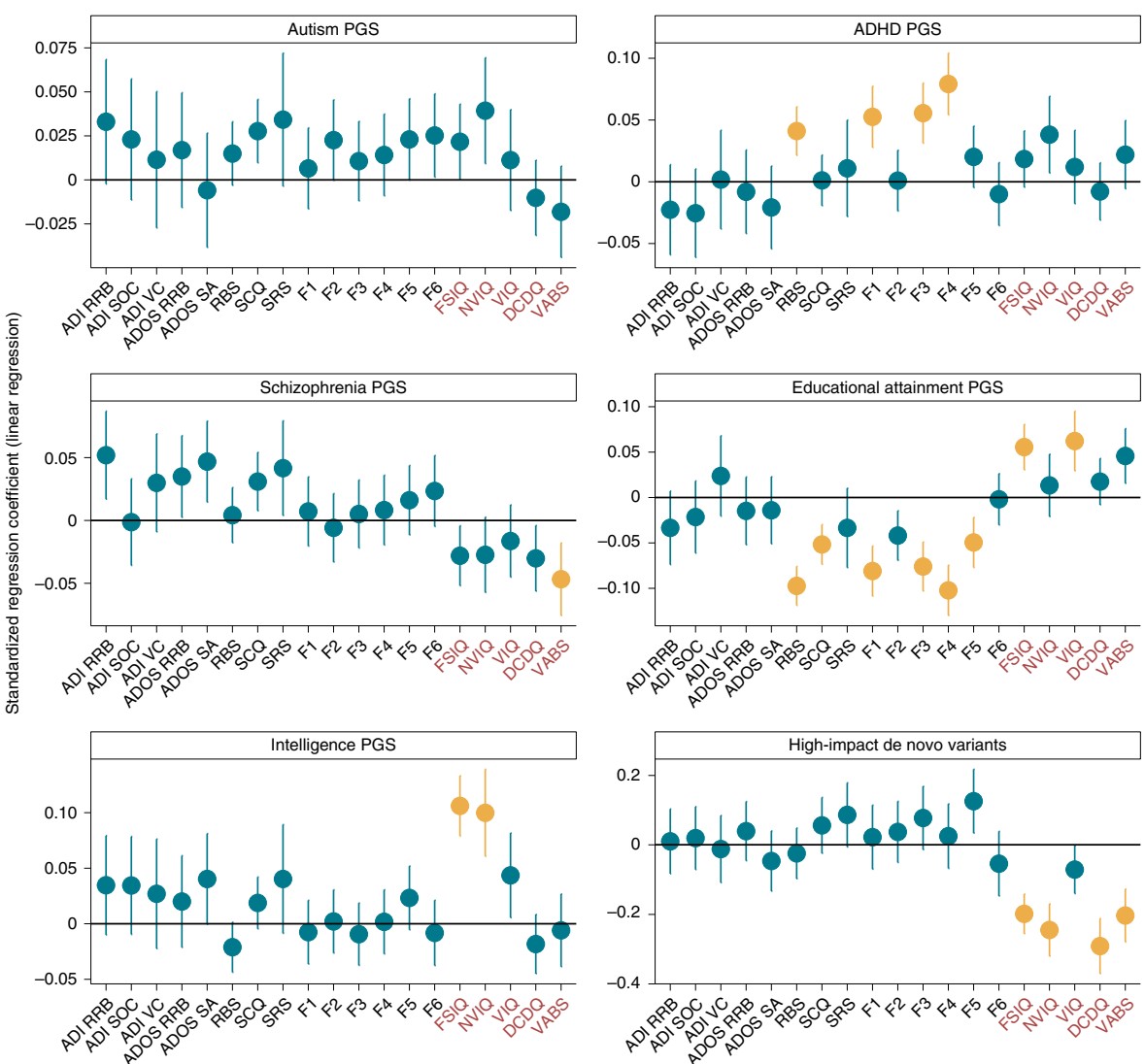

**Fig. 2 | Association of PGS and high-impact de novo variants with core and associated autism features.** Results from linear regression analyses testing the associations between the core and associated autism features and PGS for autism, ADHD, schizophrenia, educational attainment and intelligence, and with high-impact de novo variants ($n = 2,421–12,893$). For all association plots, standardized regression coefficients from linear regressions (central point) and 95% confidence intervals are provided. Yellow indicates significant association after Benjamini–Yekutieli correction for multiple comparisons (corrected $P < 0.05$). Red text indicates associated features, where higher values correspond to greater ability. Phenotypes are Autism Diagnostic Observation Schedule social affect (ADOS SA) and restricted and repetitive behavior (ADOS RRB); Autism Diagnostic Interview-Revised verbal communication (ADI VC), social interaction (ADI SOC) and restricted and repetitive behavior (ADI RRB); insistence of sameness factor (F1); social interaction factor (F2); sensory–motor behavior factor (F3); self-injurious behavior factor (F4); idiosyncratic repetitive speech and behavior factor (F5); communication skills factor (F6); adaptive behavior assessed by the Vineland Adaptive Behavior Scales (VABS); motor coordination assessed by the Development Coordination Disorder Questionnaire (DCDQ); score on the Social Responsiveness Scale (SRS); full-scale IQ (FSIQ); nonverbal IQ (NVIQ); and verbal IQ (VIQ).

autism PGS ($\beta_{PGS} = 0.028$, s.e. $= 0.045$, $P = 0.53$, logistic regression; Supplementary Fig. 6).

The PGS in a trio with an affected child can be summarized as the parental mean PGS (henceforth, midparental PGS) and the deviation of the affected child's PGS from the midparental PGS. As previously reported[14], with this expanded sample size, we identified an overtransmission of autism PGS to autistic individuals (mean $= 0.17$, s.e. $= 0.01$, $n = 6,981$, $P < 2 \times 10^{-16}$) and, curiously, a modest undertransmission to unaffected siblings (mean $= -0.03$, s.e. $= 0.02$, $n = 3,832$, $P = 0.034$) (Fig. 4b and Supplementary Table 13). This likely reflects both reproductive stoppage[41] and underdiagnosis of autism in the parental generation[42]. Carriers had a modest overtransmission of autism PGS (mean $= 0.08$, s.e. $= 0.04$, $n = 579$,

$P = 0.02$), while this was substantially higher in non-carriers (mean $= 0.18$, s.e. $0.01$, $n = 4,997$, $P < 2 \times 10^{-16}$). Notably, while carriers had significantly lower overtransmission than non-carriers ($P = 0.02$), they had a significantly higher overtransmission than siblings (PGS; $P = 9.1 \times 10^{-3}$), providing additional support for additivity of common and rare genetic variants.

A second hypothesis is that the effect of high-impact de novo variants on core autism features is partly mediated by associated autism features, given the modest negative correlation between them (Fig. 4c). Given that high-impact de novo variants are associated with a relatively sizeable reduction in both full-scale IQ and motor coordination, we reasoned that there would be a knock-on effect on core autism features. The fact that we did not observe a

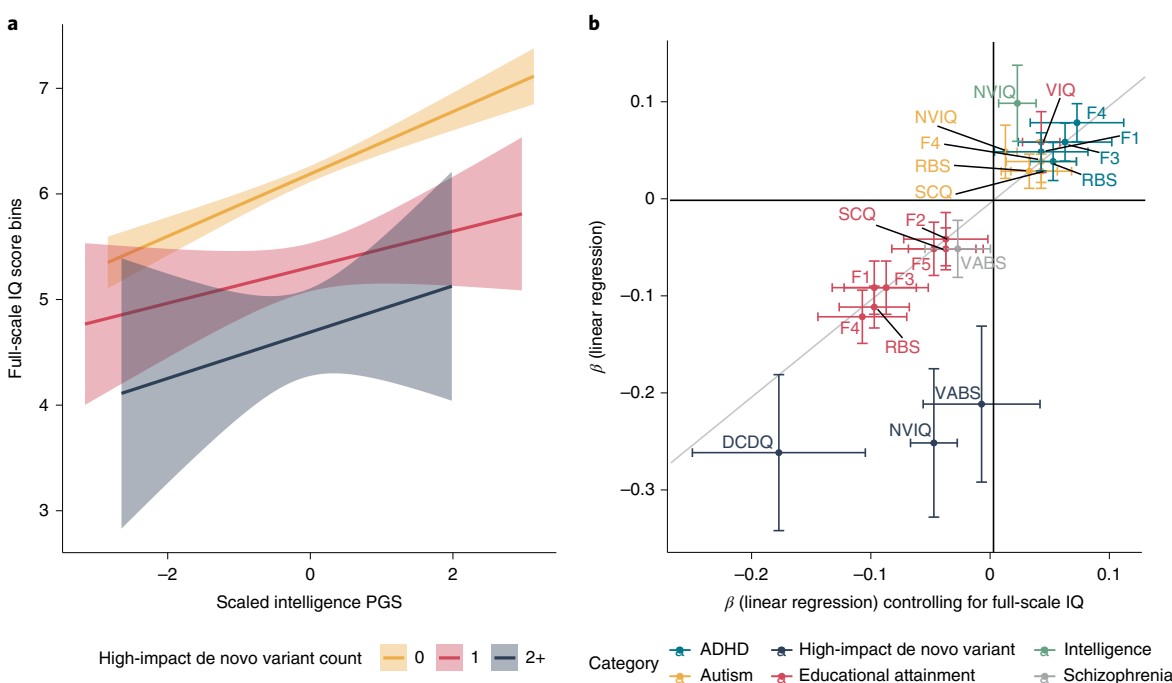

**Fig. 3 | Association between genotype and full-scale IQ and impact of full-scale IQ on genotype–phenotype associations. a**, Line plots for full-scale IQ scores as a function of intelligence PGS and counts of high-impact de novo variants in the SPARK and SSC cohorts, plotted with $n = 3,197$ autistic individuals. Only binned full-scale IQ scores were available in the SPARK cohort, and, subsequently, full-scale IQ was binned in the SSC cohort and treated as a continuous variable (Methods). Shaded regions indicate 95% confidence intervals of fitted values of the regression line. **b**, Point estimates of linear regression coefficients (central point) for the association between PGS and high-impact de novo variants and core and associated autism features without ($y$ axis) and after ($x$ axis) accounting for full-scale IQ scores ($n = 2,232$–12,893 autistic individuals). Confidence intervals (95%) for both regressions are provided. Only significant genotype–phenotype estimates are plotted. Point estimates closer to the diagonal line indicate no change in $\beta$ coefficient (linear regression) after controlling for full-scale IQ.

significant association between high-impact de novo variants and core autism features (Fig. 2b) may be due to attenuated correlations between core and associated features in carriers compared to non-carriers[21]. However, tests of matrix correlation equivalence suggested no differences in the phenotypic correlation structures of carriers and non-carriers ($P = 9.25 \times 10^{-4}$, Jennrich test for matrix equivalency). This was supported by the finding of no differences in pairwise Pearson's correlation coefficients between each of the three associated features and the six factors, SCQ and RBS between carriers and non-carriers (Fisher's $Z$-test, all $P > 0.05$).

One alternate explanation is that we are underpowered to observe this effect. We used simulations to investigate whether we had sufficient statistical power to identify associations between high-impact de novo variants and core autism features. Assuming that all effects are completely mediated by only one of the three associated features (full-scale IQ, adaptive behavior or motor coordination), power calculations indicate that we had less than 80% power for all core autism features tested (Fig. 4d). Larger samples may identify significant effects between high-impact de novo variants and core autism features, but it will be important to investigate whether the associations are mediated by associated autism features. However, neither of these two hypotheses excludes the possibility that different classes of de novo variants (for example, missense versus protein-truncating, de novo variants in specific functional categories) may be associated with core autism features.

**Autism PGS and co-occurring developmental disabilities.** Multiple co-occurring developmental disabilities are another source of heterogeneity among autistic individuals. While co-occurring developmental disabilities are associated with high-impact de novo variants[15,17,20], it is unclear whether they are impacted by PGS for

autism. In the SPARK study, in line with previous research[15,17,20], carriers of high-impact de novo variants had increased counts of co-occurring developmental disabilities ($\beta_{\text{de novo}} = 0.31$, s.e. $= 0.05$, $P = 1.55 \times 10^{-8}$, $n = 3,089$; quasi-Poisson regression). By contrast, higher PGS for autism was associated with reduced count of co-occurring developmental disabilities ($\beta_{\text{PGS}} = -0.037$, s.e. $= 0.009$, $P = 3.91 \times 10^{-5}$, $n = 13,435$, quasi-Poisson regression), even after accounting for the other three PGS (Fig. 5a and Supplementary Table 14a). Leave-one-out analyses indicated that the results were not driven by any one developmental disability (Supplementary Fig. 7). Notably, autistic individuals with five or more co-occurring developmental disabilities did not have statistically higher autism PGS than non-autistic siblings (Fig. 5a and Supplementary Table 14b). By contrast, even when restricting to autistic individuals with no co-occurring developmental disabilities, individuals with a high-impact de novo variant were more likely to be autistic than non-autistic siblings (Fig. 5a and Supplementary Table 14b).

The apparent negative association between autism PGS and co-occurring developmental disabilities has not, to our knowledge, been reported earlier. This can reflect both a true negative association (for example, PGS for autism increase IQ in both the general population[16,43] and in autistic individuals as seen in Fig. 2a) and the negative correlation between high-impact de novo variants and autism PGS. To better delineate this, we investigated the association between the two classes of genetic variants and two well-characterized developmental phenotypes: age of walking independently and age of first words. In autistic individuals, autism PGS were associated with earlier age of walking ($\beta_{\text{PGS}} = -0.012$, s.e. $= 0.003$, $P = 3.2 \times 10^{-5}$, negative binomial regression) and earlier age of first words ($\beta_{\text{PGS}} = -0.0125$, s.e. $= 0.005$, $P = 0.01$, negative binomial regression), while high-impact de novo variants increased

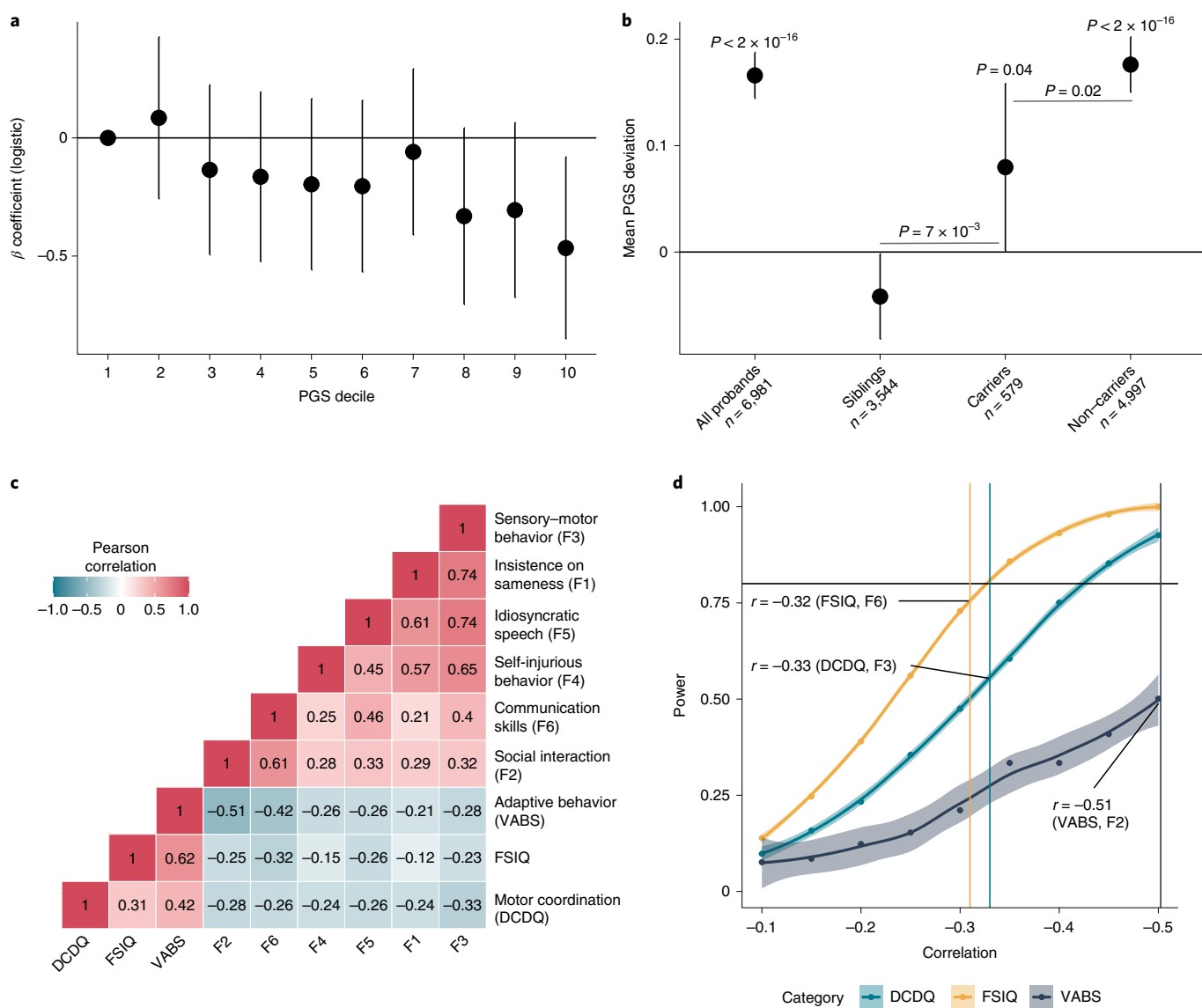

**Fig. 4 | Additivity and impact of high-impact de novo variants on core autism features. a**, β coefficients ($\beta_{de\,novo}$) and 95% confidence intervals for carrying a high-impact de novo variant per decile of autism PGS in autistic individuals after accounting for sex, age, ten genetic principal components and PGS for educational attainment, intelligence and schizophrenia, calculated using logistic regression (*n* = 5,575). **b**, Overtransmission (central point) and 95% confidence errors of PGS for autism in all probands, siblings, carriers of high-impact de novo variants and non-carriers. *P* values are provided above for the overtransmission. We also compare differences in overtransmission between carriers and non-carriers and carriers and siblings and provide the *P* values for this from two-tailed *Z*-tests. **c**, Phenotypic correlation between the core features and associated autism features. **d**, Statistical power for identifying a significant association between the number of high-impact de novo variants and core features based on the correlation with the three associated features, which is provided in **c**. The highest correlation between a core feature and an associated feature is indicated on the power graph. Shaded regions indicate 95% confidence intervals of the power curve.

the age for both phenotypes (Fig. 5b and Supplementary Table 15b). The association between autism PGS and age of walking but not age of first words remained statistically significant after accounting for high-impact de novo variants and full-scale IQ (Supplementary Table 15a). Similarly, the association between high-impact de novo variants and age of walking but not age of first words remained significant after accounting for full-scale IQ (Supplementary Table 15a). However, autism PGS were not significantly associated with either age of walking or age of first words in siblings (Supplementary Table 15a). Despite the negative association between autism PGS and the two phenotypes, even autistic individuals in the highest decile of autism PGS had higher mean age of walking and age of first words than siblings, as did autistic

non-carriers (Fig. 5b) and autistic individuals with no co-occurring developmental disability, suggesting other sources of variation in these phenotypes (Supplementary Table 15b).

There is likely heterogeneity even within the broad class of constrained genes, with differential impact on autism vis-à-vis co-occurring developmental disabilities. Previous research has attempted to disentangle this heterogeneity by comparing counts of disrupting de novo variants in autism versus those in severe developmental disorders (genetically undiagnosed developmental disorders with accompanying ID and/or developmental delays)[17]. The lack of detailed phenotypic information in the cohorts assessed renders the previous research difficult to interpret[44]. Here we take a different approach to revisit this question. Using the more detailed data

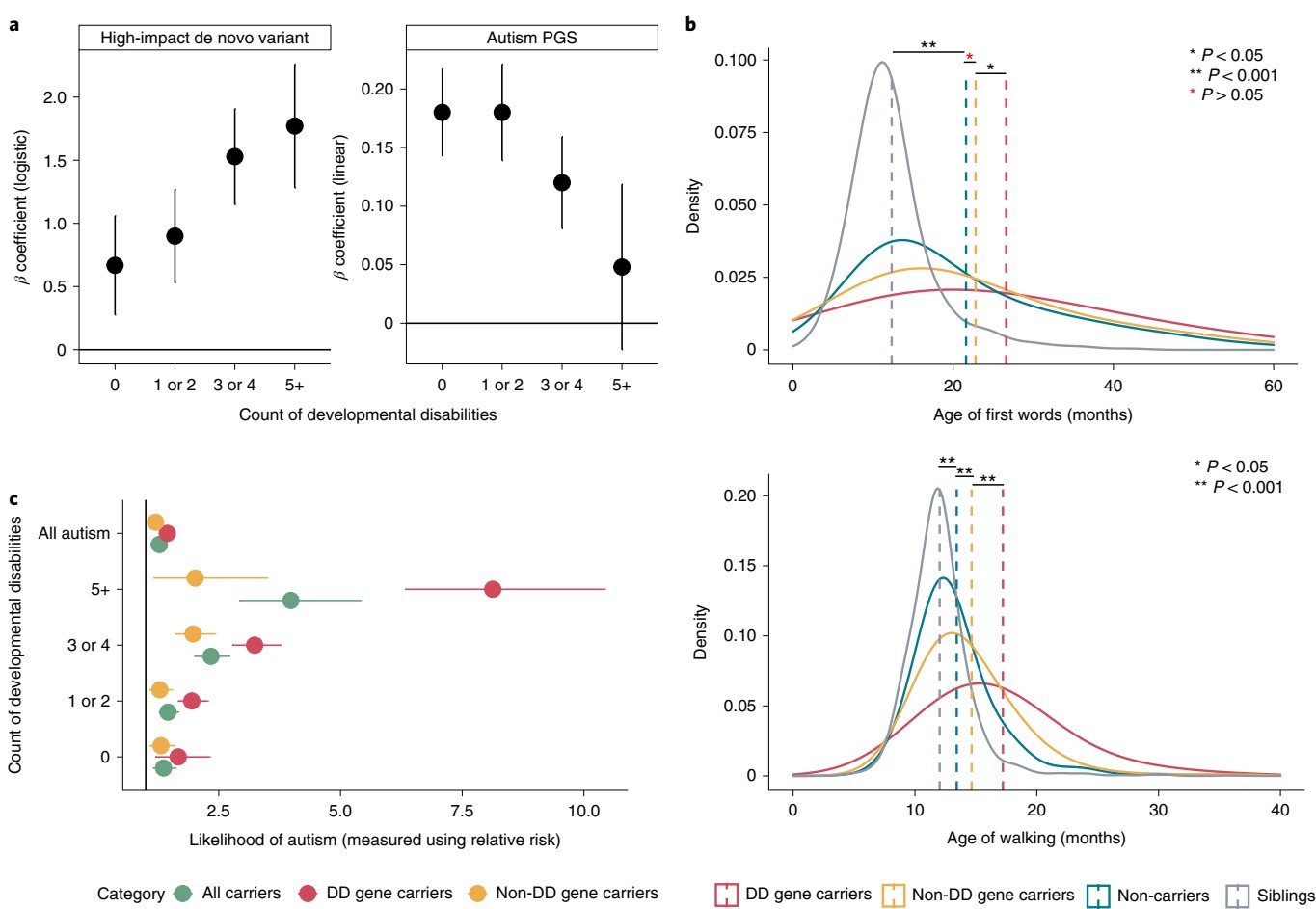

**Fig. 5 | Associations between high-impact de novo variants and autism PGS and co-occurring developmental disabilities and delays. a**, $\beta$ coefficients for the association of high-impact de novo variants (logistic regression) and autism PGS (linear regression) with case–control status (using sibling controls) by counts of co-occurring developmental disabilities. Error bars are 95% confidence intervals. **b**, Distribution and mean age of first words (top) and age of walking (bottom) in siblings, non-carriers and carriers of high-impact variants in either DD or non-DD genes. P values were calculated using Wilcoxon rank-sum tests (two-sided). **c**, Likelihood of autism, measured using relative risk, and (any number of) developmental disabilities with 95% confidence intervals for different sets of probands with high-impact de novo variants. Sibling controls were used. All relative risks were statistically significant after correcting for multiple comparisons. All data for **a,b** are from the SPARK cohort, and sample sizes are provided in Supplementary Table 13. Sample sizes for **c** are provided in Supplementary Table 14b.

on co-occurring developmental disabilities in the SPARK study, we investigated whether constrained genes robustly associated with severe developmental disorders (DD genes)[27] have differential effects on co-occurring developmental disabilities in autistic individuals compared to other constrained genes (non-DD genes). We use the term 'non-DD genes' for convenience as this list is also likely to contain genes associated with severe developmental disorders that may be discoverable at larger sample sizes but are likely less penetrant (that is, lower effect size) or lead to increased prenatal or perinatal death (that is, rarer) compared to variants in the DD genes[27].

In the SPARK cohort, 35.6% of the carriers had high-impact de novo variants in DD genes. Autistic individuals were more likely to be carriers of either set of genes than non-autistic siblings, which was observed even when restricting to autistic individuals without any known co-occurring developmental disability (Fig. 5c and Supplementary Table 14c,d). However, while the risk for the count of co-occurring developmental disabilities was elevated in carriers of DD genes ($\beta_{\text{de novo}} = 0.54$, s.e. $= 0.08$, $P = 6.48 \times 10^{-12}$; quasi-Poisson regression), this was much more modest for carriers of non-DD genes ($\beta_{\text{de novo}} = 0.15$, s.e. $= 0.07$, $P = 0.035$; quasi-Poisson regression). Supporting this, autistic carriers of high-impact de novo variants in

DD genes started walking independently and using words ~3 months later than autistic carriers of high-impact de novo variants in non-DD genes ($P < 0.05$ in both; Fig. 5b and Supplementary Table 15b). These results support a broad phenotypic distinction between the two sets of genes. We ran sensitivity analyses using a larger but overlapping list of genes identified from a highly curated database, Developmental Disorder Gene-to-Phenotype[45], and identified consistent results (Supplementary Tables 14 and 15).

**Sex differences in common and high-impact de novo variants.**
We next turned to another potential source of heterogeneity: sex. Autistic females are more likely to have high-impact de novo variants than autistic males[17,26,46,47], which is thought to support the 'female protective effect' in autism[13,46]. However, a similar effect is observed in severe developmental disorders more generally and is entirely explained by a relatively small number of genes significantly associated with severe developmental disorders (that is, DD genes)[48]. We thus revisited sex differences in high-impact de novo variants using data from the SPARK and SSC studies (Supplementary Table 16), restricting our analyses to autosomal genes. Across all high-impact de novo variants, autistic females were more likely to be carriers than

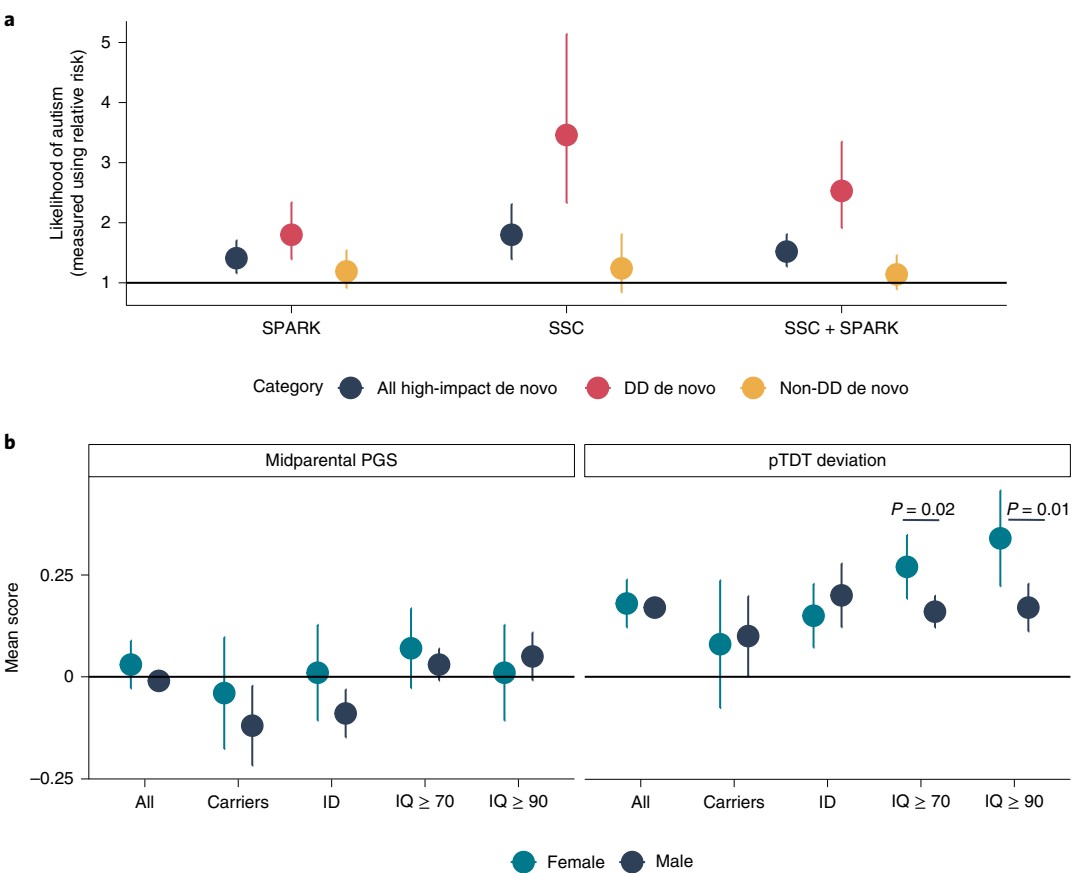

**Fig. 6 | Sex differences in high-impact de novo and common variants. a**, Likelihood for autism, measured using relative risk (central point) and 95% confidence intervals for females compared to males for being a carrier, a DD gene carrier and a non-DD gene carrier. Sample sizes are provided in Supplementary Table 15. **b**, Point estimates and 95% confidence intervals showing sex-stratified autism PGS for subgroups of autistic individuals. Left, midparental estimates. Right, overtransmitted PGS scores. All scores are standardized to midparental means. *P* values are provided from two-tailed *Z*-tests. Carriers, carriers of high-impact de novo variants; ID, autistic individuals with co-occurring ID (full-scale IQ < 70). Sample sizes are provided in Supplementary Table 17.

males (Relative risk (RR) = 1.52; 95% confidence interval, 1.27–1.81). However, this was explained entirely by high-impact de novo variants in DD genes (DD genes, RR = 2.53, 95% confidence interval = 1.91–3.35; non-DD genes, RR = 1.14, 95% confidence interval = 0.89–1.46) (Fig. 6a). This sex difference in DD genes remained and was not attenuated after accounting for the total number of co-occurring developmental disabilities in the SPARK cohort (unconditional estimates, $\beta_{\text{de novo}} = 0.83$, s.e. = 0.21, $P = 8.15 \times 10^{-5}$; conditional estimates, $\beta_{\text{de novo}} = 0.82$, s.e. = 0.22, $P = 3.53 \times 10^{-4}$; logistic regression) and after accounting for full-scale IQ and motor coordination scores in the SSC and SPARK cohorts (unconditional estimates, $\beta_{\text{de novo}} = 1.10$, s.e. = 0.15, $P = 3.42 \times 10^{-13}$; conditional estimates, $\beta_{\text{de novo}} = 1.31$, s.e. = 0.20, $P = 8.19 \times 10^{-11}$; logistic regression). We did not observe sex differences for either gene set in siblings ($P > 0.05$). These results suggest that sex differences in high-impact de novo variants are driven by a relatively small set of highly constrained genes that also increase the likelihood of co-occurring developmental disabilities in autism.

Both the contribution of PGS (Fig. 5a) and the male:female ratio are higher in autistic individuals without ID than in those with ID, suggesting that polygenic likelihood for autism may differ between sexes at IQ scores of 70 or above. Recent studies have found higher PGS for autism in females than in males[19] and greater overtransmission of PGS for autism in female non-carriers than in male carriers[49], yet neither have stratified by ID. We conducted sex-stratified polygenic transmission disequilibrium tests (pTDT) to investigate this ($n_{\text{max}} = 6,981$ autistic trios). While PGS for autism were

overtransmitted in both male and female probands, this overtransmission did not differ by sex (Fig. 6 and Supplementary Table 17). However, in autistic individuals without ID (IQ > 70), females had ~75% higher overtransmission of autism PGS than males ($P = 0.02$, two-tailed *Z*-test; Fig. 6b), which was observed even when using the sex-stratified autism genome-wide association study (GWAS) (Supplementary Table 17). When additionally removing individuals with borderline intellectual functioning (IQ < 90), females had double the overtransmission of autism PGS compared to males (females, mean = 0.34, s.e. = 0.06, $n = 276$; males, mean = 0.17, s.e. = 0.03, $n = 1,328$; difference, $P = 0.01$, two-tailed *Z*-test). We did not find any sex difference in overtransmission for autistic individuals with ID or autistic carriers of a high-impact de novo variant or non-autistic siblings. This sex difference in overtransmission was not observed for PGS for educational attainment and intelligence, suggesting that the results are not due to differences in IQ scores between sexes. Furthermore, there was no difference in midparental PGS scores, family income or parent education by sex or ID ($P > 0.05$ for all comparisons), factors correlated with participation in research[50]. This suggests that these results are unlikely to be explained by sex differences in participation. We cannot, however, distinguish the female protective effect due to common or rare variants from diagnostic bias in the current study[24,51].

**Sex and ID impact SNP heritability.** Finally, we investigated the impact of this heterogeneity on SNP heritability calculated using GREML[52,53] and phenotype correlation–genotype correlation

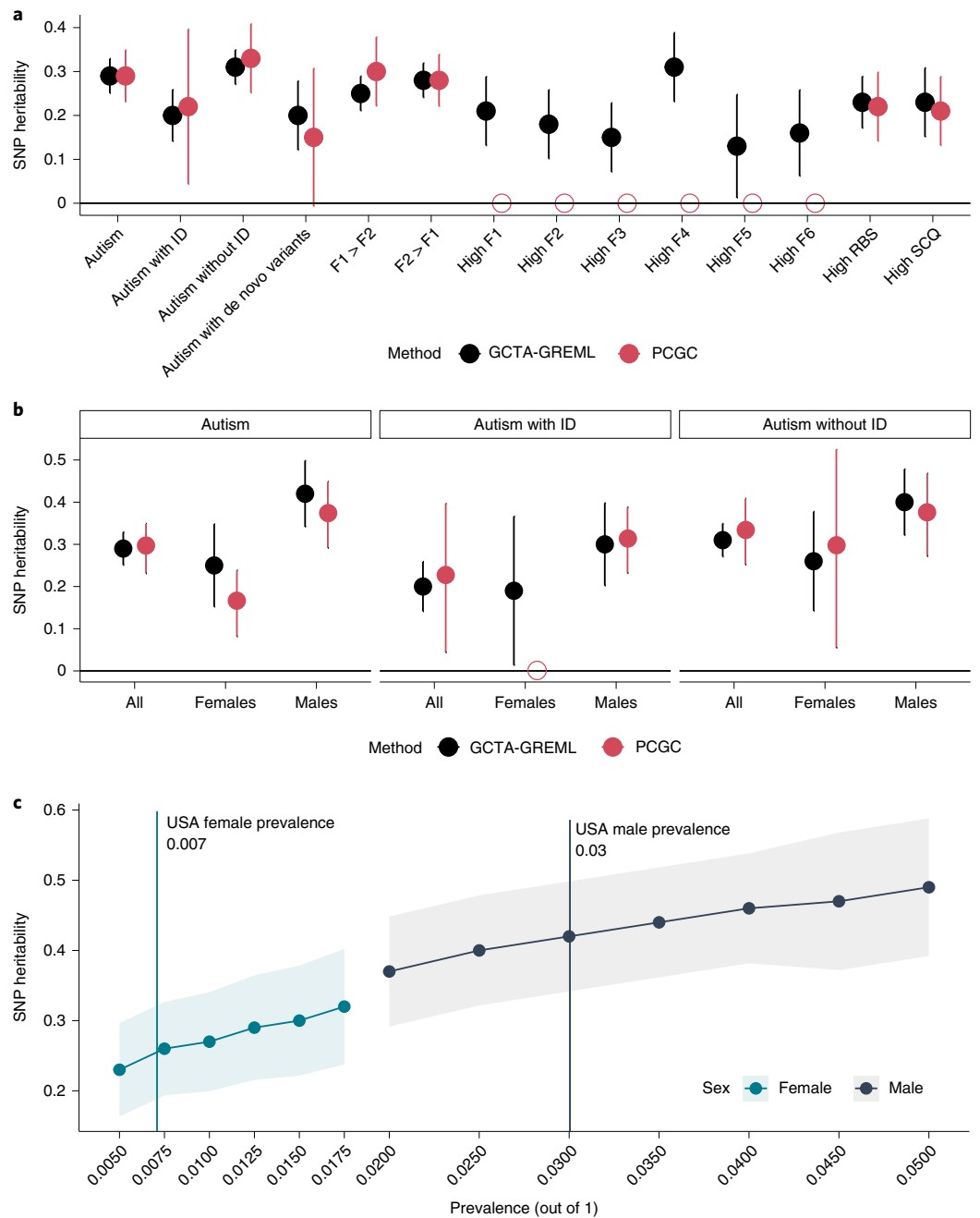

**Fig. 7 | SNP heritability estimates. a**, SNP heritability (central point) and 95% confidence intervals for various subgroups (males and females combined) of autistic individuals (maximum of $n = 4,481$ autistic individuals and 4,481 population controls). Estimates from two methods (GCTA-GREML and PCGC) are shown. Empty shapes indicate that SNP heritability was not estimated due to low statistical power. **b**, SNP heritability (central point) and 95% confidence intervals for sex- and ID-stratified autism subgroups (maximum of $n = 4,481$ autistic individuals and 4,481 population controls). Empty shapes indicate that SNP heritability was not estimated due to low statistical power. **c**, SNP heritability (central point) by sex for varying levels of autism prevalence in the USA (males, $n = 2,386$ autistic individuals and 2,386 controls; females, $n = 2,095$ autistic individuals and 2,095 controls). Shaded regions, 95% confidence intervals. The six factors are (1) insistence of sameness (F1), (2) social interaction (F2), (3) sensory–motor behavior (F3), (4) self-injurious behavior (F4), (5) idiosyncratic repetitive speech and behavior (F5) and (6) communication skills (F6).

(PCGC)[54] with individuals from the ABCD cohort as population controls (Methods). All heritability estimates are reported on the liability scale (Fig. 7a and Supplementary Table 18).

We identified a modest SNP heritability for autism (GCTA, $h^2_{SNP} = 0.29$, s.e. = 0.02; PCGC, $h^2_{SNP} = 0.29$, s.e. = 0.03), which is higher than estimates from iPSYCH[16] but lower than estimates from the AGRE[55] and PAGES[56] cohorts. Autistic individuals with ID had lower SNP heritability than autistic individuals without ID

($P = 1.6 \times 10^{-3}$, two-tailed $Z$-test). SNP heritability for autism in autistic carriers compared to general population controls (agnostic of carrier status) was modest (GCTA, $h^2_{SNP} = 0.20$, s.e. = 0.05; PCGC, $h^2_{SNP} = 0.14$, s.e = 0.08), which is similar to the SNP heritability observed for autistic individuals with ID. However, when comparing autistic high-impact de novo carriers with autistic non-carriers, the SNP heritability was not statistically significant (GCTA, $h^2_{SNP} = 0.14$, s.e. = 0.14; PCGC, $h^2_{SNP} = 0.15$, s.e. = 0.19), suggesting

that the observed SNP heritability for autistic carriers reflects autism rather than factors associated with the generation of germline mutations[57,58]. This result is in line with our pTDT analyses, which identify an overtransmission of PGS in carriers, and previous research that has identified a smaller yet significant heritability for severe developmental disorders[59].

Stratifying by sex had the largest effect on SNP heritability (Fig. 7b). Males had approximately 70% higher SNP heritability than females ($P = 9.3 \times 10^{-3}$, two-tailed $Z$-test). This difference was observed across a range of prevalence estimates (Fig. 7c and Supplementary Table 19) after downsampling the number of autistic males to match the number of autistic females (Supplementary Table 18) and varying the male:female ratio to 3.3:1 to account for diagnostic bias[51] (Supplementary Table 18). By contrast, stratifying individuals by high scores (1 s.d. above the mean) on the core autism phenotypes or a combination of two core autism phenotypes modestly reduced or did not alter the SNP heritability for autism (Fig. 7a and Supplementary Table 18).

## Discussion

Individual differences among autistic individuals in core and associated features are complex and genetically multifactorial. High-impact de novo variants and PGS have differential and often independent effects on these features. There is additivity between common and high-impact de novo variants in autism. These represent the most widely studied class of genetic variants in autism thus far, yet emerging evidence suggests a role for other classes (for example, rare inherited and de novo tandem repeats) of genetic variants as well[17,19,60,61]. However, this negative correlation between high-impact de novo variants and autism PGS may not extend to the general population. Because we have focused only on autistic individuals and not the general population, we may have induced a negative correlation between them because people have to have either a high PGS or high-impact de novo variants to cross the diagnostic threshold.

The two classes of genetic variants do not have the same effects on either the core or associated autism phenotypes nor on co-occurring developmental disabilities. The negative association between autism PGS and co-occurring developmental disabilities reflects both a true negative association (for example, for IQ[43]) and the additivity between rare and common variants.

We observe sizeable differences in both common and high-impact de novo variants based on sex and ID. While these results may be interpreted as providing support for the female protective effect[13,46], this interpretation is not straightforward. First, the increased likelihood of being a carrier of high-impact de novo variants was observed only with genes associated with severe developmental disorders, not for other constrained genes, despite both sets of genes increasing the likelihood for autism. This suggests that the female protective effect may be for severe developmental disorders rather than for autism specifically, which warrants further investigation. Second, the higher overtransmission of autism PGS must be interpreted alongside the reduced SNP heritability of autism in females. Assuming high genetic correlation between males and females, reduced SNP heritability in females suggests that higher PGS are required to reach the equivalent levels of genetic likelihood in males[62]. Yet this raises another important question: why do autistic females have lower SNP heritability than autistic males? Does this reflect ascertainment bias in the GWAS cohorts, diagnostic bias, diagnostic overshadowing, camouflaging or masking and/ or social stigma[7,24,51]? Several social factors can influence diagnosis in a sex-differential manner, and investigating this is paramount to understanding sex-differential genetic effects.

In conclusion, our findings have important implications for using genetics to understand autism. We need deeper phenotyping at scale and need to account for the evolving diagnostic criteria for autism[63].

## Online content

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

## EU-AIMS LEAP

**Antonia San Jose Caceres[22], Hannah Hayward[22], Daisy Crawley[22], Jessica Faulkner[22], Jessica Sabet[22], Claire Ellis[22], Bethany Oakley[22], Eva Loth[22], Tony Charman[22], Declan Murphy[22], Rosemary Holt[1], Jack Waldman[1], Jessica Upadhyay[1], Nicola Gunby[1], Meng-Chuan Lai[1], Gwilym Renouf[1], Amber Ruigrok[1], Emily Taylor[1], Hisham Ziauddeen[1], Julia Deakin[1], Simon Baron-Cohen[1,40], Sara Ambrosino di Bruttopilo[23], Sarai van Dijk[23], Yvonne Rijks[23], Tabitha Koops[23], Miriam Douma[23], Alyssia Spaan[23], Iris Selten[23], Maarten Steffers[23], Anna Ver Loren van Themaat[23], Nico Bast[24], Sarah Baumeister[24], Larry O'Dwyer[25], Carsten Bours[25], Annika Rausch[25], Daniel von Rhein[25], Ineke Cornelissen[25], Yvette de Bruin[25], Maartje Graauwmans[25], Elzbieta Kostrzewa[26], Elodie Cauvet[26], Kristiina Tammimies[26], Rouslan Sitnikow[26], Guillaume Dumas[7], Yang-Min Kim[7] and Thomas Bourgeron[7,40]**

[22]King's College London, London, UK. [23]UMC Utrecht, Utrecht, the Netherlands. [24]Central Institute of Mental Health Mannheim, Mannheim, Germany. [25]Radboud University Medical Centre, Nijmegen, the Netherlands. [26]Karolinska Institutet, Solna, Sweden. [40]These authors jointly supervised this work: Thomas Bourgeron, Simon Baron-Cohen.

**iPSYCH-Autism Working Group**

**David M. Hougaard[8,27], Jonas Bybjerg-Grauholm[8,27], Thomas Werge[8,28], Preben Bo Mortensen[8,29], Ole Mors[8,30] and Merete Nordentoft[8,31]**

[27]Center for Neonatal Screening, Department for Congenital Disorders, Statens Serum Institut, Copenhagen, Denmark. [28]Institute of Biological Psychiatry, MHC Sct Hans, Mental Health Services Copenhagen, Roskilde, Denmark and Department of Clinical Medicine, University of Copenhagen, Copenhagen, Denmark. [29]Center for Genomics and Personalized Medicine, National Centre for Register-Based Research, Aarhus University, and Centre for Integrated Register-based Research, Aarhus University, Aarhus, Denmark. [30]Psychosis Research Unit, Aarhus University Hospital, Aarhus, Denmark. [31]Mental Health Services in the Capital Region of Denmark, Mental Health Center Copenhagen, University of Copenhagen, Copenhagen, Denmark.

**Spectrum 10K and APEX Consortia**

**Varun Warrier[1], Dwaipayan Adhya[1], Armandina Alamanza[1], Carrie Allison[1], Isabelle Garvey[1], Rosemary Holt[1], Tracey Parsons[1], Paula Smith[1], Alex Tsompanidis[1], Simon Baron-Cohen[1,40], Graham J. Burton[32], Alexander E. P. Heazell[33], Lidia V. Gabis[34,35], Tal Biron-Shental[35,36], Madeline A. Lancaster[37], Deepak P. Srivastava[38] and Jonathan Mill[39]**

[32]Centre for Trophoblast Research, University of Cambridge, Cambridge, UK. [33]Obstetrics and Director Tommy's Maternal and Fetal Research Centre, the University of Manchester, Manchester, UK. [34]Child Development Division, Maccabi Health Services, Tel Aviv, Israel. [35]Sackler Faculty of Medicine, Tel Aviv University, Tel Aviv, Israel. [36]Department of Obstetrics and Gynecology, Meir Medical Center, Kefar Sava, Israel. [37]MRC Laboratory of Molecular Biology, University of Cambridge, Cambridge, UK. [38]MRC Centre for Neurodevelopmental Disorders, King's College London, London, UK. [39]University of Exeter Medical School, College of Medicine & Health, University of Exeter, Exeter, UK.

## Methods

**Participants.** For factor analyses, we restricted our analyses to autistic individuals from the SSC and SPARK cohorts. Participants had to have completed the two phenotypic measures (details are below) to be included in the factor analyses. We also excluded autistic individuals with incomplete entries in either of the two measures ($n = 5,754$ only in the SPARK cohort). This resulted in 1,803 participants ($n = 1,554$ males) in the SSC, 14,346 participants ($n = 11,440$ males) in SPARK version 3 and 8,271 participants ($n = 6,262$ males) in extra entries from SPARK version 5 (SSC, mean age = 108.75 months, s.d. = 43.29 months; SPARK version 3, mean age = 112.11 months, s.d. = 46.43 months; SPARK version 5, mean age = 111.22 months, s.d. = 48.19 months). Only the SCQ was available for siblings in the SPARK study.

We conducted analyses using data from four cohorts of autistic individuals: the SSC ($n = 8,813$)[30], the Autism Genetic Resource Exchange (AGRE, CHOP sample) ($n_{max} = 1,200$)[64], the AIMS-2-TRIALS Longitudinal European Autism Project (LEAP) sample ($n_{max} = 262$)[65] and SPARK ($n = 29,782$)[31]. For sibling comparisons, we included siblings from the SSC ($n = 1,829$) and SPARK ($n = 12,260$) cohorts. For trio-based analyses, we restricted to complete trios in the SSC ($n = 2,234$) and SPARK ($n = 4,747$) cohorts. For all analyses, we restricted the sample to autistic individuals who passed genetic quality control (QC) and who had phenotypic information.

**Factor analyses.** *Phenotypes.* We conducted factor analyses using the SCQ[29] and the RBS[28]. The SCQ is a widely used caregiver report of autistic traits capturing primarily social communication difficulties and, to a lesser extent, repetitive and restricted behaviors[29]. There are 40 binary (yes-or-no) questions in total, with the first question focusing on the individual's ability to use phrases or sentences (total score, 0–39). We used the Lifetime version rather than the current version as this was available in both the SPARK and SSC studies. Of note, in the Lifetime version, questions 1–19 are about behavior over the lifetime, while questions 20–40 refer to behavior between the ages of 4 to 5 years or in the last 12 months if the participant is younger. We excluded participants who could not communicate using phrases or sentences ($n = 217$ in the SSC and $n = 17,092$ in SPARK) as other questions in the SCQ were not applicable to this group of participants. The RBS is a caregiver-reported measure of presence and severity of repetitive behaviors over the last 12 months. It consists of 43 questions assessed on a four-point Likert scale (total score, 0–129). Higher scores on both measures indicate greater autistic traits.

*Exploratory factor analyses.* We conducted exploratory factor analysis on a random half of the SSC ($n = 901$ individuals, of which 782 were males) using 'promax' rotation to identify correlated factors as implemented by 'psych' (ref. [66]) in R. We conducted three sets of exploratory correlated factor analyses: for all items, for social items and for non-social items. Previous studies have provided support for a broad dissociation between social and non-social autism features[12,23] and have conducted separate factor analyses of social (for example, refs. [67,68]) and non-social autism features (for example, refs. [69,70]). Thus, we reasoned that separating items into social and non-social categories might aid the identification of covariance structures that may not be apparent when analyzing all items together. We divided the data into social (all of the SCQ except item 1 and nine other items and item 28 from the RBS) and non-social (nine items from the SCQ (items 8, 11, 12 and 14–18) and all items from RBS except item 28) items, which was carried out after discussion between V.W. and X.Z. The ideal number of factors to be extracted was identified from examining the scree plot (Supplementary Fig. 2), parallel analyses and theoretical interpretability of the extracted factors. However, we examined all potential models using confirmatory factor analyses as well to obtain fit indices, and the final model was identified using both exploratory and confirmatory factor analyses.

We then applied the model configurations from 'promax' rotated exploratory factor analysis for bifactor models to explore the existence of general factor(s). In addition to a single general factor bifactor model, we divided the data into social and non-social items as mentioned earlier and applied bifactor models separately for the social and non-social items. Hierarchical $\Omega$ values and explained common variances were then calculated for potential models as extra indicators of the feasibility of bifactor models, but hierarchical $\Omega$ values were not greater than 0.8 for most of the models tested, and explained common variances were not greater than 0.7 (refs. [71–73]) for any of the models tested (Supplementary Table 2).

*Confirmatory factor analyses.* Three rounds of confirmatory factor analyses were conducted: first for the second half of the SSC, followed by analysis of SPARK participants whose phenotypic data were available in version 3 of the data release and, finally, analysis of SPARK participants whose phenotypic data were available only in version 4 or version 5 of the data release and not in the earlier releases. To evaluate the models, multiple widely adopted fit indices were considered, including the comparative fit index (CFI), the TLI and the root mean square error of approximation. In CFA, items are assigned only to the factor with the highest loading to attain parsimony. We conducted three broad sets of confirmatory factor analyses: (1) confirmatory factor analyses of all correlated factor models, (2) confirmatory factor analyses of the autism bifactor model and (3) confirmatory factor analyses of social and non-social bifactor models. For

each of these confirmatory factor models, we limited the number of factors tested based on the slope of the scree plots and based on the number of items loading onto the factor (five or more). For the confirmatory factor analyses of social and non-social bifactor models, we iteratively combined various numbers of social and non-social group factors. In bifactor models, items without loading onto the general factor in the correspondent EFA were excluded. Items were allocated to different group factors, which were identified based on the highest loading (items with loading <0.3 were excluded). Due to the ordinal nature of the data, all CFAs were conducted using the diagonally weighted least-squares estimator (to account for the ordinal nature of the data) in the R package lavaan 0.6-5 (ref. [74]). We identified the model most appropriate for the data at hand with TLI and CFI > 0.9 (TLI and CFI > 0.95 for bifactor models), low root mean square error of approximation and good theoretical interpretability based on discussions between V.W. and X.Z. Additionally, as sensitivity analyses, the identified model (correlated six-factor model) was run again with two orthogonal method factors mapping onto SCQ and RBS-R to investigate if the fit indices remained high after accounting for covariance between items derived from the same measure, as these measures vary subtly during the period of time evaluated. We also reanalyzed the identified model after removing items that were loaded onto multiple factors (>0.3 on two or more factors) to provide clearer theoretical interpretation of the model. For genetic analyses, we used factor scores from the correlated six-factor model without including the orthogonal method factors and without dropping the multi-loaded items.

**Genetic analyses.** *Genetic quality control.* QC was conducted for each cohort separately by array. We excluded participants with genotyping rate <95%, excessive heterozygosity (±3 s.d. from the mean), non-European ancestry as detailed below, mismatched genetic and reported sex and, for families, those with Mendelian errors >10%. SNPs with genotyping rate <10% were excluded, or they were excluded if they deviated from Hardy–Weinberg equilibrium ($P < 1 \times 10^{-6}$). Given the ancestral diversity in the SPARK cohort, Hardy–Weinberg equilibrium and heterozygosity were calculated in each genetically homogeneous population separately. Genetically homogeneous populations (corresponding to five super-populations: African, East Asian, South Asian, admixed American and European) were identified using the five genetic principal components calculated using SPARK and 1000 Genomes Phase 3 populations[75] and clustered using UMAP[76]. Principal components were calculated using linkage disequilibrium-pruned SNPs ($r^2 = 0.1$, window size = 1,000 kb, step size = 500 variants, after removing regions with complex linkage disequilibrium patterns) using GENESIS[77], which accounts for relatedness between individuals, calculated using KING[78].

Imputation was conducted using the Michigan Imputation Server[79] with 1000 Genomes phase 3 version 5 as the reference panel[49] (for AGRE and SSC), with the HRC r1.1 2016 reference panel[80] (for AIMS-2-TRIALS) or using the TOPMed imputation panel[81] (for both releases of SPARK). Details of imputation have been previously reported[82]. SNPs were excluded from polygenic risk scores if they had minor allele frequency <1%, had an imputation $r^2 < 0.4$ or were multi-allelic.

*Polygenic scores.* We restricted our PGS associations to four GWAS. First, we included a GWAS of autism from the latest release from the iPSYCH cohort (iPSYCH-2015)[83]. This includes 19,870 autistic individuals (15,025 males and 4,845 females) and 39,078 individuals without an autism diagnosis (19,763 males and 19,315 females). All individuals included in this GWAS were born between May 1980 and December 2008 to mothers who were living in Denmark. GWAS was conducted on individuals of European ancestry, with the first ten genetic principal components included as covariates using logistic regression as provided in PLINK. Further details are provided elsewhere[49]. We additionally included GWAS for educational attainment ($n = 766,345$, excluding the 23andMe dataset)[35], intelligence ($n = 269,867$)[34], ADHD ($n = 20,183$ individuals diagnosed with ADHD and 35,191 controls)[36] and schizophrenia (69,369 individuals diagnosed with schizophrenia and 236,642 controls)[37]. These GWAS were selected given the relatively large sample size and modest genetic correlation with autism. Additionally, as a negative control, we included PGS generated from a GWAS of hair color (blonde versus other, $n = 43,319$ blondes and $n = 342,284$ others) from the UK Biobank, which was downloaded from https://atlas.ctglab.nl/traitDB/3495. This phenotype has SNP heritability comparable to that of the other GWAS used ($h^2 = 0.15$, s.e. = 0.014), is unlikely to be genetically or phenotypically correlated with autism and related traits, and has a sample size large enough to be a reasonably well-powered negative control.

PGS were generated for three phenotypes using polygenic risk scoring with continuous shrinkage (PRS-CS)[84], which is among the best-performing polygenic scoring methods using summary statistics in terms of variance explained[85]. In addition, this method bypasses the step of identifying a P-value threshold. We set the global shrinkage prior ($\varphi$) to 0.01, as is recommended for highly polygenic traits. Details of the SNPs included are provided in Supplementary Table 3.

De novo variants were obtained from Antaki et al.[19]. De novo variants (structural variants and SNVs) were called for all SSC samples and a subset of the SPARK samples (phase 1 genotype release, SNVs only). To identify high-impact de novo SNVs, we restricted data to variants with a known effect on protein.

These are damaging variants: 'transcript_ablation', 'splice_acceptor_variant', 'splice_donor_variant', 'stop_gained', 'frameshift_variant', 'stop_loss', 'start_loss' or missense variants with MPC[86] scores >2. We further restricted data to variants in constrained genes with a LOEUF score <0.37 (ref. [87]), which represent the topmost decile of constrained genes. For SVs, we restricted data to SVs affecting the most constrained genes, that is, those with LOEUF score <0.37, representing the first decile of most constrained genes. We did not make a distinction between deletions or duplications. To identify carriers, non-carriers and parents, we restricted our data to samples from the SPARK and SSC studies that had been exome sequenced and families in which both parents and the autistic proband(s) passed the genotyping QC.

For genes associated with severe developmental disorders, we obtained the list of constrained genes that are significant genes associated with severe developmental disorders from Kaplanis et al.[27]. To investigate the association of this set of genes with autism and developmental disorders, we first identified autistic carriers with a high-impact de novo variant and then divided this group into carriers who had at least one high-impact de novo variant in a DD gene and carriers with high-impact de novo variants in other constrained genes.

Only individuals with undiagnosed developmental disorders are recruited into the Deciphering Developmental Disorders study, and, as such, known genes associated with developmental disorders that are easy for clinicians to recognize and diagnose may be omitted from the genes identified by Kaplanis et al.[27]. To account for this bias, we ran sensitivity analyses using a larger but overlapping list of genes identified from the Developmental Disorder Gene-to-Phenotype database (DDG2P). From this database, we used constrained genes that are either 'confirmed' or 'probable' developmental disorder genes and genes for which heterozygous variants lead to developmental phenotypes (that is, mono-allelic or X-linked dominant).

**Phenotypes.** *Core and associated autism features.* We identified 19 autism core and associated features that (1) are widely used in studies related to autism; (2) are a combination of parent-, self- and other-reported and performance-based measures to investigate if reporter status affects the PGS association; (3) are collected in all three cohorts; and (4) cover a range of core and associated features in autism. The core features are

1. ADOS[88]: social affect
2. ADOS[88]: restricted and repetitive behavior domain total score
3. ADI[89]: communication (verbal) domain total score
4. ADI[89]: restricted and repetitive behavior domain total score
5. ADI[89]: social domain total score
6. RBS[28]
7. Parent-reported Social Responsiveness Scale-2 (ref. [90]): total raw scores
8. SCQ[29]
9. Insistence of sameness factor (F1)
10. Social interaction factor (F2)
11. Sensory–motor behavior factor (F3)
12. Self-injurious behavior factor (F4)
13. Idiosyncratic repetitive speech and behavior (F5)
14. Communication skills factor (F6).

The associated features are

1. Vineland Adaptive Behavior Scales[91]: composite standard scores
2. Full-scale IQ
3. Verbal IQ
4. Nonverbal IQ
5. Developmental Coordination Disorders Questionnaire[92].

Measures of IQ were quantified using multiple methods across the range of IQ scores in the AGRE, SSC and LEAP studies. In the SPARK study, IQ scores were available based on parent reports on ten IQ score bins (Fig. 1c). We used these as full-scale scores. For analyses involving the SPARK and SSC cohorts, we converted full-scale scores from the SSC into IQ bins to match what was available from the SPARK study and treated them as continuous variables based on examination of the frequency histogram (Supplementary Fig. 8). For the six factors, we excluded individuals who were minimally verbal (Factor analyses), but these individuals were not excluded for analyses with other autism features.

*Developmental phenotypes.* We identified seven questions relating to developmental delay in the SPARK medical screening questionnaire. These are all binary questions (yes or no). Summed scores ranged from 0 to 7. The developmental phenotypes include the presence of

1. ID, cognitive impairment, global developmental delay or borderline intellectual functioning
2. Language delay or language disorder
3. Learning disability (learning disorder, including reading, written expression or math; or nonverbal learning disability)
4. Motor delay (for example, delay in walking) or developmental coordination disorder
5. Mutism

6. Social (pragmatic) communication disorder (as included in DSM IV TR and earlier)
7. Speech articulation problems.

We included the age of first words and the age of walking independently for further analyses. This was recorded using parent-reported questionnaires in the SPARK study and in ADI-R[89] in the SSC study. While other developmental phenotypes are available, we focused on these two, as they represent major milestones in motor and language development and are relatively well characterized.

**Statistics.** *Note of distribution of phenotypes and statistical analyses.* Before any statistical analyses, we visually inspected the distributions of the variables. All continuous variables were approximately normally distributed with the exception of the 'age of first words', the 'age of walking independently' and the count of co-occurring developmental disabilities. For these three variables, we used quasi-Poisson or negative binomial regression to account for overdispersion in the data and because the variance was much greater than the mean. These models produced the same estimate but modestly different standard errors. Both have two parameters. However, while quasi-Poisson regression models the variance as a linear function of the mean, the negative binomial models the variance as a quadratic function of the mean. The model that produced the lower residual deviance was chosen between the two. For all other continuous variables, we used linear regression and parametric tests. For binary data, we used logistic regression as there was not a large imbalance in the case:control ratio.

*Genetic association analyses.* For each cohort, PGS and high-impact de novo variants were regressed against the autism features with sex and the first ten genetic principal components as covariates in all analyses, with all continuous independent variables standardized. In addition, array was included as a covariate in SSC and AGRE datasets. This was performed using linear regression for standardized quantitative phenotypes, logistic regression for binary phenotypes (for example, association between PGS and the presence of a high-impact de novo variant), Poisson regression for count data (number of developmental disorders or delays, not standardized) and negative binomial regression for the age of walking independently or the age of first words (not standardized; MASS[93] package in R).

For the association between genetic variables and core and associated autism phenotypes, we first conducted linear regression analyses for the four PGS first using multivariate regression analyses with data from SPARK (waves 1 and 2), SSC, AGRE and AIMS-2-TRIALS LEAP. This is of the form:

$$y \approx \text{PGS}_{\text{autism}} + \text{PGS}_{\text{schizophrenia}} + \text{PGS}_{\text{EA}} + \text{PGS}_{\text{intelligence}} + \text{sex} + \text{age} + 10\text{PCs},$$
(1)

where EA is educational attainment and 10PCs are ten principal components. For the negative control, we added the negative control as an additional independent variable in equation (1):

$$y \approx \text{PGS}_{\text{autism}} + \text{PGS}_{\text{schizophrenia}} + \text{PGS}_{\text{EA}} + \text{PGS}_{\text{intelligence}}$$
$$+ \text{PGS}_{\text{hair color}} + \text{sex} + \text{age} + 10\text{PCs}.$$
(2)

For the AGRE and SPARK studies, we ran equivalent mixed-effects models with family ID modeled as random intercepts to account for relatedness between individuals. This was carried out using the lme4 (ref. [94]) package in R.

For high-impact de novo variants, we included the count of high-impact de novo variants as an additional independent variable in equation (1) and ran regression analyses for SPARK (wave 1 only) and SSC. To ensure interpretability across analyses, we retained only individuals who passed the genotypic QC, which included only individuals of European ancestries. Family ID was included as a random intercept:

$$y \approx \text{PGS}_{\text{autism}} + \text{PGS}_{\text{schizophrenia}} + \text{PGS}_{\text{EA}} + \text{PGS}_{\text{intelligence}}$$
$$+ \text{high-impact de novo count} + \text{sex} + \text{age} + 10\text{PCs}.$$
(3)

Effect sizes were meta-analyzed across the three cohorts using inverse-variance-weighted meta-analyses with the following formula:

$$w_i = \text{SE}_i^{-2}$$
$$\text{SE}_{\text{meta}} = \sqrt{\left( (\Sigma_1 w_i)^{-1} \right)}$$
$$\beta_{\text{meta}} = \Sigma_i \beta_i w_i (\Sigma_i w_i)^{-1},$$
(4)

where $\beta_i$ is the standardized regression coefficient of the PGS, $\text{SE}_i$ is the associated standard error and $w_i$ is the weight. $P$ values were calculated from $Z$ scores. Given the high correlation between the autism features and phenotypes, we used Benjamini–Yekutieli false discovery rates to correct for multiple testing (corrected $P < 0.05$). We calculated heterogeneity statistics (Cochran's $Q$ and $I^2$ values) for the PGS meta-analyses but not for the associations with high-impact de novo variants, as the latter were calculated using only two datasets (SSC and SPARK).

For the SPARK and SSC studies, we investigated the association between PGS (equation (1)) and being a carrier of a high-impact de novo variant (equation (3)) and the age of first walking and first words using negative binomial regression and conducted inverse-variance meta-analyses (equation (4)). We ran the same analyses for the SPARK study to investigate the association between PGS (equation (1)) and high-impact de novo variants (equation (3)) and counts of co-occurring developmental disabilities (quasi-Poisson regression). Leave-one-out analyses were conducted by systematically excluding one of seven co-occurring developmental disabilities and reconducting the analyses.

To investigate additivity between common and high-impact de novo variants, we conducted logistic regression with carrier status as a dependent binary variable and all PGS included as independent variables and genetic principal components, sex and age included as covariates. This was carried out separately for SPARK (wave 1) and SSC and meta-analyzed as outlined earlier.

*Phenotypic analyses.* Statistical significance of differences in factor scores between sexes were computed using *t*-tests. Associations with age and IQ bins were conducted using linear regressions after including sex as a covariate.

Matrix equivalency tests were conducted using the Jennrich test in the psych[66] package in R. Power calculations were conducted using simulations. Statistical differences between pairwise correlation coefficients (carriers versus non-carriers) in core and associated features were tested using the package cocor[95] in R. Using scaled existing data on full-scale IQ, adaptive behavior and motor coordination, we generated correlated simulated variables at a range of correlation coefficients to reflect the correlation between the six core factors and the three associated features. We then ran regression analyses using the simulated variable and high-impact de novo variants as provided in equation (3). We repeated this 1,000 times and counted the fraction of outcomes for which the association between high-impact de novo variant count and the simulated variable had $P < 0.05$ to obtain statistical power. Differences in the age of walking and the age of first words between groups of autistic individuals and siblings were calculated using Wilcoxon rank-sum tests.

*Sex differences: polygenic transmission disequilibrium tests.* Polygenic transmission deviation was conducted using polygenic transmission disequilibrium tests[14]. To allow comparisons with midparental scores, residuals of the autism PGS were obtained after regressing out the first ten genetic principal components. These residuals were standardized by using the parental mean and standard deviations. We obtained similar results using PGS that had not been residualized for the first ten genetic principal components. We defined individuals without co-occurring ID as individuals whose full-scale IQ is above 70 the SSC and SPARK studies. Additionally, in the SPARK cohort, we excluded any of these participants who had a co-occurring diagnosis of 'intellectual disability, cognitive impairment, global developmental delay or borderline intellectual functioning'. Analyses were conducted separately in the SSC and SPARK cohorts and meta-analyzed using inverse-variance-weighted meta-analyses. We additionally conducted pTDT analyses on non-autistic siblings to investigate differences between males and females.

*Sex differences: high-impact de novo variants.* For sex differences in high-impact de novo variants, we calculated relative risk in autistic females versus males based on (1) all carriers, (2) carriers of DD genes and (3) carriers of non-DD genes (SPARK wave 1 and SSC). For sensitivity analyses, we conducted logistic regression with sex as the dependent variable and carrier status for DD genes and either full-scale IQ and motor coordination scores (in SPARK wave 1 and SSC) or number of developmental disorders (only in SPARK wave 1) as covariates. For each sensitivity analysis, we provide the estimates of the unconditional analysis as well (that is, without the covariates).

**Heritability analyses.** We opted to conduct heritability analyses using unscreened population controls rather than family controls (that is, pseudocontrols or unaffected family members), as this likely reduces SNP heritability[96] owing to parents having higher genetic likelihood for autism compared to unselected population controls[55] and due to assortative mating[97]. Case–control heritability analyses were conducted using the ABCD cohort as population controls; specifically, the ABCD child cohort in the USA, recruited at the age of 9 or 10 years. This cohort is reasonably representative of the US population in terms of demographics and ancestry. As such, it represents an excellent comparison cohort for the SPARK and SSC cohorts. The ABCD cohort was genotyped using the Smokescreen genotype array, a bespoke array designed for the study containing over 300,000 SNPs. Genetic QC was conducted identically as for SPARK. Genetically homogeneous groups were identified using the first five genetic principal components followed by UMAP clustering with the 1000 Genomes data. We restricted our analyses to 4,481 individuals of non-Finnish European ancestries in the ABCD cohort. Scripts for this are available at https://github.com/vwarrier/ABCD_geneticQC. Imputation was conducted, similar to the analysis of SPARK data, using the TOPMed imputation panel.

For case–control heritability analyses, we combined genotype data from the ABCD cohort and from autistic individuals from the SPARK and SSC cohorts. We restricted the analysis to 6,328,651 well-imputed SNPs ($r^2 > 0.9$) with minor allele frequency >1% in all datasets. Furthermore, we excluded multi-allelic

SNPs and SNPs with minor allele frequency difference of >5% between the three datasets and, in the combined dataset, were not in Hardy–Weinberg equilibrium ($P > 1 \times 10^{-6}$) or had genotyping rate <99%. We additionally excluded related individuals, identified using GCTA-GREML, and individuals with genotyping rate <95%. We calculated genetic principal components for the combined dataset using 52,007 SNPs with minimal linkage disequilibrium ($r^2 = 0.1$, 1,000 kb, step size of 500 variants, removing regions with complex long-range linkage disequilibrium). Visual inspection of the principal-component plots did not identify any outliers (Supplementary Fig. 9). While our QC procedure is stringent, we note that there will be unaccounted-for effects in SNP heritability due to fine-scale population stratification, differences in genotyping array and participation bias in the autism cohorts. However, our focus is on the differences in SNP heritability between subgroups of autistic individuals, and unaccounted-for case–control differences will not affect this.

We calculated SNP heritability for autism and additionally in subgroups stratified for the presence of ID, sex, sex and ID together, and the presence of high-impact de novo variants. We also conducted SNP heritability in subgroups of autistic individuals with scores >1 s.d. from the mean for each of the six factors, autistic individuals with F1 scores > F2 scores and autistic individuals with F2 scores > F1 scores.

We calculated the observed-scale SNP heritability (baseline and subgroups) using GCTA-GREML[52,53] and, additionally, using PCGC[54]. In all models except for the sex-stratified models, we included sex, age in months and the first ten genetic principal components as covariates. In the sex-stratified models, we included age in months and the first ten genetic principal components as covariates. For sex-stratified heritability analyses, both case and control data were from the same sex. For GCTA-GREML, the observed-scale SNP heritability was converted into liability-scale SNP heritability using equation (23) from Lee et al.[98]. PCGC estimates SNP heritability directly on the liability scale using the prevalence rates from Maenner et al.[99]. For all analyses, we ensured that the number of cases did not exceed the number of controls, with a maximum case:control ratio of 1.

We used prevalence rates from Maenner et al.[99], which provides prevalence of autism among 8 year olds (1.8%). The study also provides prevalence rates by sex and by the presence of ID. However, there is wide variation in autism prevalence. We thus recalculated the SNP heritability across a range of state-specific prevalence estimates obtained from Maenner et al.[99]. For estimates of liability-scale heritability for subtypes defined by factor scores >1 s.d. from the mean, we estimated a prevalence of 16% of the total prevalence. For F1 > F2 and F2 > F1, prevalence was estimated at 50% of the total autism prevalence. Estimating approximate population prevalence of autistic individuals with high-impact de novo variant carriers is difficult due to ascertainment bias in existing autism cohorts. However, a previous study has demonstrated that the mutation rate for rare protein-truncating variants is similar between autistic individuals and siblings from the SSC and autistic individuals and population controls from the iPSYCH sample in Denmark, which does not have a participation bias[100], implying that the de novo mutation rate in autistic individuals from the SPARK and SSC cohorts may be generalizable. Using the sex-specific proportion of de novo variant carriers and autism prevalence, we calculated a prevalence of 0.2% for being an autistic carrier of a high-impact de novo variant.

For sex-stratified SNP heritability analyses, we additionally calculated SNP heritability for a range of state-specific prevalence estimates to better model state-specific factors that contribute to autism diagnosis. In addition, using a total prevalence of 1.8%, we estimated SNP heritability using a male:female ratio of 3.3:1 (ref. [51]) to account for diagnostic bias that may inflate the ratio.

We used GCTA-GREML to also estimate SNP heritability for the six factors, full-scale IQ and the bivariate genetic correlation between them. We used the same set of SNPs used in the case–control analyses. We were unable to conduct bivariate genetic correlation for the case–control datasets due to limitations of sample size.

**Ethics.** We received ethical approval to access and analyze de-identified genetic and phenotypic data from the three cohorts from the University of Cambridge Human Biology Research Ethics Committee.

**Reporting summary.** Further information on research design is available in the Nature Research Reporting Summary linked to this article.

## Data availability

Genetic and phenotypic data for SFARI and SPARK are available upon application and approval from the Simons Foundation (https://www.sfari.org/resource/autism-cohorts/). Approved researchers can obtain the SPARK and SSC population datasets described in this study by applying at https://base.sfari.org. Data for AGRE are available upon application and approval from Autism Speaks (https://www.autismspeaks.org/agre). Data for EU-AIMSLEAP are available upon application and approval to the EU-AIMSLEAP committee (https://www.eu-aims.eu/the-leap-study). DDG2P phenotypes can be obtained here: https://www.deciphergenomics.org/ddd/ddgenes. GWAS data are available for hair color (https://atlas.ctglab.nl/traitDB/3495), schizophrenia and ADHD (https://www.med.unc.edu/pgc/download-results/), intelligence (https://ctg.cncr.nl/software/summary_statistics/) and educational attainment (https://thessgac.com/).

## Code availability

All scripts used in this study are available as follows: genetic QC and imputation in SSC (https://github.com/vwarrier/SSC_liftover_imputation; basic scripts used for imputing the SSC genotyped datasets), genetic QC and imputation in SPARK (https://github.com/vwarrier/SPARK_QC_imputation; QC and imputation of the SPARK dataset), genetic QC and imputation in the ABCD cohort (https://github.com/vwarrier/ABCD_geneticQC), bespoke genetic analyses (https://github.com/vwarrier/autism_heterogeneity; this git has the code for the heterogeneity in the Autism Project). We used the following software packages: PRScs (https://github.com/getian107/PRScs; polygenic prediction via continuous shrinkage priors), the TOPMed Imputation Server (https://imputation.biodatacatalyst.nhlbi.nih.gov/), PLINK (PLINK 2.0 (https://www.cog-genomics.org/)), GCTA-GREML (PLINK 2.0 (https://cnsgenomics.com/)), PCGC (PCGC regression (https://dougspeed.com/)). The following R packages were used: psych 2.1.6, cocor 1.1-3, lavaan 0.6-5, MASS 7.3-54, lme4 1.1-27.1.

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

## Acknowledgements

S.B.-C. received funding from the Wellcome Trust (214322\Z\18\Z). For the purpose of open access, we have applied a CC BY public copyright licence to any author accepted manuscript version arising from this submission. S.B.-C. also received funding from the Autism Centre of Excellence, the SFARI, the Templeton World Charitable Fund, the MRC and the National Institute for Health Research Cambridge Biomedical Research Centre. The research was supported by the National Institute for Health Research Applied Research Collaboration East of England. Any views expressed are those of the author(s) and not necessarily those of the funder. Some of the results leading to this publication have received funding from the Innovative Medicines Initiative 2 Joint Undertaking under grant agreement no. 777394 for the project AIMS-2-TRIALS. This joint undertaking receives support from the European Union's Horizon 2020 research and innovation program and the EFPIA and Autism Speaks, Autistica and the SFARI. V.W. is funded by St. Catharine's College, Cambridge. T.B. has received funding from the Institut Pasteur, the CNRS, the Bettencourt–Schueller and the Cognacq–Jay Foundations, the APHP and the Université de Paris Cité. We acknowledge with gratitude the generous support of D. and M. Gillings in strengthening the collaboration between S.B.-C. and T.B. and between Cambridge University and the Institut Pasteur. The iPSYCH team was supported by grants from the Lundbeck Foundation (R102-A9118, R155-2014-1724 and R248-2017-2003), the NIMH (1U01MH109514-01 to A.D.B.) and the universities and university hospitals of Aarhus and Copenhagen. The Danish National Biobank resource was supported by the Novo Nordisk Foundation. High-performance computer capacity for handling and statistical analysis of iPSYCH data on the GenomeDK HPC facility was provided by the Center for Genomics and Personalized Medicine and the Centre for Integrative Sequencing, iSEQ, Aarhus University, Denmark (grant to A.D.B.). We thank J. Sebat for sharing the de novo variant calls in the SPARK and SSC datasets. We are grateful to all families at the participating SSC sites as well as the principal investigators (A. Beaudet, R. Bernier, J. Constantino, E. Cook, E. Fombonne, D. Geschwind, R. Goin-Kochel, E. Hanson, D. Grice, A. Klin, D. Ledbetter, C. Lord, C. Martin, D. Martin, R. Maxim, J. Miles, O. Ousley, K. Pelphrey, B. Peterson, J. Piggot, C. Saulnier, M. State, W. Stone, J. Sutcliffe, C. Walsh, Z. Warren and E. Wijsman). We are grateful to all families in the SPARK study, the SPARK clinical sites and SPARK staff.

## Author contributions

V.W., T.B. and S.B.-C. conceived the study. V.W., X.Z., P.R., C.S.L., T.R., A. Rosengren, F.C. and J.G. performed analyses. T.M.M. provided statistical input on factor analyses.

A.H. provided input on clinical interpretation of factor scores and co-occurring developmental conditions. H.C.M. and J.G. provided input on other statistical analyses. A.D.B and J.G. provided summary statistics for the iPSYCH autism dataset. D.H.R., M.E.H., D.H.G., A.D.B. and E.B.R. provided input on interpreting the results. V.W. wrote the manuscript with the help of all authors.

## Competing interests

M.E.H. is a cofounder of and consultant to and holds shares in Congenica Ltd., a genetics diagnostics company.

## Additional information

**Correspondence and requests for materials** should be addressed to Varun Warrier or Simon Baron-Cohen.

# Reporting Summary

## Statistics

For all statistical analyses, confirm that the following items are present in the figure legend, table legend, main text, or Methods section.

| n/a | Confirmed | |
|---|---|---|
| ☐ | ☒ | The exact sample size (*n*) for each experimental group/condition, given as a discrete number and unit of measurement |
| ☐ | ☒ | A statement on whether measurements were taken from distinct samples or whether the same sample was measured repeatedly |
| ☐ | ☒ | The statistical test(s) used AND whether they are one- or two-sided<br>*Only common tests should be described solely by name; describe more complex techniques in the Methods section.* |
| ☐ | ☒ | A description of all covariates tested |
| ☐ | ☒ | A description of any assumptions or corrections, such as tests of normality and adjustment for multiple comparisons |
| ☐ | ☒ | A full description of the statistical parameters including central tendency (e.g. means) or other basic estimates (e.g. regression coefficient) AND variation (e.g. standard deviation) or associated estimates of uncertainty (e.g. confidence intervals) |
| ☐ | ☒ | For null hypothesis testing, the test statistic (e.g. *F*, *t*, *r*) with confidence intervals, effect sizes, degrees of freedom and *P* value noted<br>*Give P values as exact values whenever suitable.* |
| ☒ | ☐ | For Bayesian analysis, information on the choice of priors and Markov chain Monte Carlo settings |
| ☒ | ☐ | For hierarchical and complex designs, identification of the appropriate level for tests and full reporting of outcomes |
| ☐ | ☒ | Estimates of effect sizes (e.g. Cohen's *d*, Pearson's *r*), indicating how they were calculated |

*Our web collection on statistics for biologists contains articles on many of the points above.*

## Software and code

Policy information about availability of computer code

| Data collection | No software has been used for data collection |
|---|---|

| Data analysis | ● Genetic QC and imputation in SSC: vwarrier/SSC_liftover_imputation: Basic scripts used for imputing the SSC genotyped datasets (github.com)
● Genetic QC and imputation in SPARK: vwarrier/SPARK_QC_imputation: QC and imputation of the SPARK dataset (github.com)
● Genetic QC and imputation in ABCD: vwarrier/ABCD_geneticQC (github.com)
● Bespoke genetic analyses: vwarrier/autism_heterogeneity: This git has the code for the heterogeneity in autism project (github.com)
We used the following software packages:
● PRScs: getian107/PRScs: Polygenic prediction via continuous shrinkage priors (github.com); March 4, 2021 version
● TOPMED imputation server: TOPMed Imputation Server (nih.gov)
● Plink: PLINK 2.0 (cog-genomics.org)
● GCTA-GREML: GCTA v 1.93.3 beta(cnsgenomics.com)
● PCGC: PCGC Regression | dougspeed.com; LDAK 5.1 linux

R packages:
1. Psych 2.1.6
2. Cocor 1.1-3
3. Lavaan 0.6-5
4. MASS 7.3-54
5. lme4 1.1-27.1 |

For manuscripts utilizing custom algorithms or software that are central to the research but not yet described in published literature, software must be made available to editors and reviewers. We strongly encourage code deposition in a community repository (e.g. GitHub). See the Nature Portfolio guidelines for submitting code & software for further information.

## Data

Policy information about availability of data

All manuscripts must include a data availability statement. This statement should provide the following information, where applicable:

- Accession codes, unique identifiers, or web links for publicly available datasets
- A description of any restrictions on data availability
- For clinical datasets or third party data, please ensure that the statement adheres to our policy

Genetic and phenotypic data for SFARI and SPARK are available upon application and approval from the Simons Foundation (SFARI | Autism Cohorts). Approved researchers can obtain the SPARK and SSC population dataset described in this study by applying at https://base.sfari.org. Data for AGRE is available upon application and approval from Autism Speaks (AGRE - Autism Genetic Resource Exchange | Autism Speaks). Data for EU-AIMS Leap is available upon application and approval to the EU-AIMS LEAP committee (The LEAP Study (eu-aims.eu)). Ddg2p phenotype can be obtained here: DECIPHER v11.9: Mapping the clinical genome (deciphergenomics.org). GWAS data availability: Hair colour (https://atlas.ctglab.nl/traitDB/3495; Schizophrenia and ADHD (Download Results | Psychiatric Genomics Consortium (unc.edu)); intelligence (GWAS Summary Statistics | CTG (cncr.nl)); educational attainment (SSGAC Login (thessgac.com)).

# Field-specific reporting

Please select the one below that is the best fit for your research. If you are not sure, read the appropriate sections before making your selection.

☒ Life sciences          ☐ Behavioural & social sciences          ☐ Ecological, evolutionary & environmental sciences

For a reference copy of the document with all sections, see nature.com/documents/nr-reporting-summary-flat.pdf

# Life sciences study design

All studies must disclose on these points even when the disclosure is negative.

| Sample size | We used the largest available sample size, combining data from four different cohorts. |
| Data exclusions | We excluded individuals who did not pass genetic quality control. |
| Replication | No direct replication was conducted. We conducted meta-analyses of all available data, and used orthogonal methods to validate the results. |
| Randomization | Randomization was only needed for factor analyses. This was done in R. No randomization was needed for other analyses. For association analyses using polygenic scores and SNP heritability analyses we included sex, age and genetic principal components. |
| Blinding | No blinding was conducted in this study as autistic individuals and non-autistic individuals were identified by diagnosis. |

# Reporting for specific materials, systems and methods

We require information from authors about some types of materials, experimental systems and methods used in many studies. Here, indicate whether each material, system or method listed is relevant to your study. If you are not sure if a list item applies to your research, read the appropriate section before selecting a response.

## Materials & experimental systems

| n/a | Involved in the study |
|---|---|
| ☒ | ☐ Antibodies |
| ☒ | ☐ Eukaryotic cell lines |
| ☒ | ☐ Palaeontology and archaeology |
| ☒ | ☐ Animals and other organisms |
| ☐ | ☒ Human research participants |
| ☒ | ☐ Clinical data |
| ☒ | ☐ Dual use research of concern |

## Methods

| n/a | Involved in the study |
|---|---|
| ☒ | ☐ ChIP-seq |
| ☒ | ☐ Flow cytometry |
| ☒ | ☐ MRI-based neuroimaging |

## Human research participants

Policy information about studies involving human research participants

| | |
|---|---|
| Population characteristics | For factor analyses, we restricted our analyses to autistic individuals from the Simons Simplex Collection (SSC) and SPARK cohorts. Participants had to have completed the two phenotypic measures (details below) to be included in the factor analyses. We also excluded autistic individuals with incomplete entries in either of the two measures (N = 5,754 only in SPARK). This resulted in 1,803 participants (N = 1,554 males) in SSC, 14,346 (N = 11,440 males) in SPARK version 3 and 8,271 (N = 6,262 males) in extra entries from SPARK version 5 (SSC: Mean age = 108.75, SD = 43.29 ; SPARK version 3: Mean age = 112.11 months, SD = 46.43; SPARK version 5: Mean age = 111.22 months, SD = 48.19). Only the SCQ was available for siblings in SPARK. We conducted analyses using data from four cohorts of autistic individuals: The Simons Simplex Collection (SSC, N = 8,813)[30], the Autism Genetic Resource Exchange (AGRE, CHOP sample) (Nmax = 1,200)[64], the AIMS-2-TRIALS LEAP sample (Nmax = 262)[65], and SPARK (N = 29,782)[31]. For sibling comparisons, we included siblings from SSC (N = 1,829) and SPARK (N = 12,260). For trio-based analyses, we restricted to complete trios in SSC (N = 2,234) and SPARK (N = 4,747). For all analyses we restricted the sample to autistic individuals who passed genetic quality control and who had phenotypic information. |
| Recruitment | Participants were recruited through clinics, labs, and online. This was done by others. |
| Ethics oversight | University of Cambridge's Human Biology Research Ethics Committee provided ethical approval to analyse pseudonymised phenotypic and genetic data. |

Note that full information on the approval of the study protocol must also be provided in the manuscript.

