## [Peer Review File. · Nature Genetics]

Peer Review Information

Manuscript Title: Genetic correlates of phenotypic heterogeneity in autism

Corresponding author name(s): Dr Varun Warriar

Reviewer Comments & Decisions:

Decision Letter, initial version:
--

17th September 2021

Dear Varun,

Your Article "Genetic correlates of phenotypic heterogeneity in autism" has been seen by two referees. You will see from their comments below that, while they find your work of interest, they have raised several relevant points. We are interested in the possibility of publishing your study in Nature Genetics, but we would like to consider your response to these points in the form of a revised manuscript before we make a final decision on publication.

To guide the scope of the revisions, the editors discuss the referee reports in detail within the team, including with the chief editor, with a view to identifying key priorities that should be addressed in revision, and sometimes overruling referee requests that are deemed beyond the scope of the current study. In this case, we particularly ask that you address all technical queries related to the statistical analyses and their interpretation and extend the analyses where feasible to address potential biases inherent to the study design. We hope you will find this prioritized set of referee points to be useful when revising your study. Please do not hesitate to get in touch if you would like to discuss these issues further.

We therefore invite you to revise your manuscript taking into account all reviewer and editor comments. Please highlight all changes in the manuscript text file. At this stage we will need you to upload a copy of the manuscript in MS Word .docx or similar editable format.

*2) If you have not done so already please begin to revise your manuscript so that it conforms to our Article format instructions, available

[here](http://www.nature.com/ng/authors/article_types/index.html).

*3) Include a revised version of any required Reporting Summary:

[REDACTED]

We hope to receive your revised manuscript within 8-12 weeks. If you cannot send it within this time, please let us know.

Sincerely,
Kyle

Kyle Vogan, PhD
Senior Editor
Nature Genetics
<https://orcid.org/0000-0001-9565-9665>

Referee expertise:

Referee #1: Genetics, neuropsychiatric disorders

Referee #2: Genetics, neuropsychiatric disorders

Reviewers' Comments:

Reviewer #1:
Remarks to the Author:

Warrier and colleagues conducted a carefully designed study to investigate the effect of common and rare variation in the autism phenotypic spectrum. I found the methods generally adequate and the results interesting. However, there are few issues that I would like the authors to consider.

1. The several heritabilities and genetic correlations reported in Supplementary Tables 5 and 6 are not significantly different from 0. Accordingly, in the main text, it should be clearly reported which features can be interpreted from a genetic point of view.
2. The authors used different regression models. It should be clearly reported which assumptions the authors made on the distribution of the variables to select the most appropriate model for each analysis. Additionally, the authors used Pearson's correlation for the correlation reported in Figure 1A. Since this is a parametric test, did the authors verify the normal distribution of the variables tested?
3. In addition to ASD PGS, the authors investigated also other PGS. It's surprising that ADHD PGS was not included in the analysis. The PGC cross-disorder analysis showed that genetic liabilities to ADHD and ASD converge on the same latent factor (PMID:31835028). Because of known overlapping features and this convergent genetic component, the authors should consider including ADHD PGS in their analysis.
4. Since the authors have access to the individual data of the dataset used to generate the ASD PGS, they can verify whether there is a sample overlap due to shared or related participants.
5. It appears that the sex-stratified analysis was conducted using PGS derived from GWAS not stratified by sex. This could likely introduce biases related to the sex distribution of the GWAS used to generate the PGS. The authors should stratify the iPSYCH-2015 by sex and generate female- and male-specific effects to generate their PGS. Additionally, more information should be provided regarding the age and sex characteristics of the iPSYCH-2015 cohort.

6. The authors observed sex differences in the effect of high impact de novo variants, especially in those affecting DD genes. Did the authors verify whether this effect may be driven by X-linked loci?

7. The authors meta-analyzed the effects estimated across multiple cohorts. It would be important to report the heterogeneity estimates related to the meta-analyses conducted.

Reviewer #2:

Remarks to the Author:

With great interest I have reviewed the manuscript 'Genetic correlates of phenotypic heterogeneity in autism'. In their manuscript, the authors provide valuable new insights that allow us to understand genetic heterogeneity in autism. The manuscript covers a broad scope of analyses of both common and rare genetic variation in relation to autism. The analyses are sound and presented very well in the figures, and the manuscript is well-written. The analyses support their main message that a dimensional approach to autism aids in understanding its highly heterogeneous factors, which is a step forward from the heterogeneous case definitions that have been used in GWAS.

A review of the manuscript has left me with a couple of questions and remarks:

Results: Figure 1: panels B to D have 'mean score' on the y-axis. I find it difficult to interpret the effect size in these figures. I assume these are standardized effects? Would there be a way to put these mean score differences into perspective, i.e. give the reader a better idea of the magnitude of the observed effects?

Results: the authors report lower factor scores in older participants which may relate to recall bias of parent-reported scores in younger individuals. To what extent does selection bias play a role here? Since older participants often provide consent themselves (in contrast to children participants where parents consent), it may well be those adults with less severe symptoms that are willing/able to participate in a research study.

Results: it would be of interest to know whether their dimensional approach actually changes gene finding efforts. This would be easy to do (but may be beyond the scope) by carrying out genome-wide association analyses and compare gene-mapping results/genetic correlations with the regular autism GWAS that is published.

Results: PRS analyses: it is not quite clear from the text what discovery GWAS was used to obtain effect sizes for the PRS of autism (and schizophrenia, intelligence), please provide reference/sample size.

Results: the lack of association between de novo variants and the autism scales is somewhat surprising. Might this relate to a selection effect as well? (i.e. lack of cases with highly penetrant/pathogenic mutations in the cohort). In any case, since these do not associate with autism dimensions it seems somewhat confusing to conclude that correcting PRS analyses for these non-significant effects means the PRS effects are independent.

Results: Figure 2: the effect sizes of autism PRS with autism symptoms are rather small (assuming

these are standardized effect sizes in a linear regression model). Could the authors comment on this, would this relate to the limited explained variance by PRS based on the latest GWAS?

Results: Figure 3A: why did the authors chose to bin IQ on the y-axis? (in unequal bins)

Results: the observation of a lower PRS in carrier autistic individuals compared to non-carriers is an interesting finding. How much of this may relate to collider bias between the two?

Author Rebuttal to Initial comments

Reviewers' Comments:

Reviewer #1:

Remarks to the Author:

Warrier and colleagues conducted a carefully designed study to investigate the effect of common and rare variation in the autism phenotypic spectrum. I found the methods generally adequate and the results interesting. However, there are few issues that I would like the authors to consider.

We thank the reviewer for this kind assessment of the manuscript. We have endeavoured to address the comments below.

1.1. The several heritabilities and genetic correlations reported in Supplementary Tables 5 and 6 are not significantly different from 0. Accordingly, in the main text, it should be clearly reported which features can be interpreted from a genetic point of view.

We agree. We have now revised our Results (fourth paragraph under Identifying latent phenotypes in core autism features) to read as follows:

“Of the six factors and RBS and SCQ, only Insistence on Sameness (F1) and Self-Injurious Behaviour (F4) had significant SNP heritability, which was similar to that of full-scale IQ measured in autistic individuals (**Supplementary Table 5**).”

1.2. The authors used different regression models. It should be clearly reported which assumptions the authors made on the distribution of the variables to select the most appropriate model for each analysis. Additionally, the authors used Pearson's correlation for the correlation reported in Figure 1A. Since this is a parametric test, did the authors verify the normal distribution of the variables tested?

For all tests conducted, we visually inspected the variables. Where the variables appeared to be distributed approximately normally (which included the six factors), we

conducted linear regression. We used logistic regression for binary variables - note, none of the binary variables had a high-imbalance in case:control ratio to warrant using Firth logistic regression. For the count of co-occurring disabilities and developmental milestones (age at first words and age of walking independently, both measured in months) we used quasi-poisson and negative binomial respectively as the data were overdispersed, and the variance was much greater than the mean. Both negative binomial and quasi-poisson are two-parameter models that model the variance as a function of the mean (linear for quasi-poisson and quadratic for negative binomial). There is no established convention for choosing between the two, and they both produce identical estimates but modestly differing standard errors. Ultimately, we used the model that produced lower residual deviance for the data at hand.

We have added the following note in the Methods to clarify this.

“Note of distribution of phenotypes and statistical analyses

Prior to any statistical analyses, we visually inspected the distribution of the variables. All continuous variables were approximately normally distributed with the exception of ‘age of first words’ and ‘age of walking independently’, and count of co-occurring developmental disabilities. For these three variables, we used quasi-poisson or negative binomial regression, to account for overdispersion in the data, and because the variance was much greater than the mean. These models produce the same estimate but modestly different standard errors. Both have two parameters. However, whilst the quasi-poisson models the variance as a linear function of the mean, the negative binomial models the variance as a quadratic function of the mean. The model that produced the lower residual deviance was chosen between the two. For all other continuous variables, we used linear regression and parametric tests. For binary data, we used logistic regression as there was not a large imbalance in case-control ratio.”

1.3. In addition to ASD PGS, the authors investigated also other PGS. It's surprising that ADHD PGS was not included in the analysis. The PGC cross-disorder analysis showed that genetic liabilities to ADHD and ASD converge on the same latent factor (PMID:31835028). Because of known overlapping features and this convergent genetic component, the authors should consider including ADHD PGS in their analysis.

We agree. We have now conducted analyses using the latest ADHD PGS, and updated the Results, Methods, and Supplementary Tables. We have reproduced the edited paragraph from the Results below:

“We first investigated the association between the 19 features and PGS (**Methods**) for autism (iPSYCH autism data-freeze, **Methods**), intelligence³⁴, educational attainment³⁵, ADHD³⁶, and schizophrenia³⁷, and, as a negative control, hair colour³⁸ (N = 2,421 to 12,893, sample sizes in **Supplementary Table 7**). In multiple regression analyses, ADHD PGS^{1#} were associated with increased non-social core autism features (total scores on the RBS, insistence on sameness, sensory-motor, and self-injurious factor scores) (**Figure 2A, Supplementary Table 8**). Intelligence PGS were associated with increased full-scale and non-verbal IQ. Educational attainment PGS were associated with increased full-scale and verbal IQ and reduced scores on core autism features. Schizophrenia PGS were associated with reduced adaptive behaviour, measured using the composite score of the Vineland Adaptive Behaviour Scales. Finally, after multiple testing, autism PGS were not significantly associated with any of the features tested, possibly due to reduced statistical power, but was nominally associated with both increased measures of IQ and social core autism features. Moderate heterogeneity ($I^2 > 50\%$) was observed only for 10% of the associations. The majority of the significant associations (12 out of 15) had concordant effect directions in all cohorts (**Supplementary Figure 5**). We did not identify any significant genotype-phenotype association using hair colour (blonde vs other) as a negative control (**Supplementary Table 8**).”

1.4. Since the authors have access to the individual data of the dataset used to generate the ASD PGS, they can verify whether there is a sample overlap due to shared or related participants.

Unfortunately, we do not have access to individual level data for the iPSYCH dataset. This is because of fairly strict data access to individual level data from iPSYCH (a Danish cohort). It is highly unlikely that there is any overlap that will meaningfully affect the data as three of the four major cohorts are from the USA (SSC, SPARK, and AGRE), and EU-AIMS LEAP does not include participants from Denmark.

^{1#} Throughout the manuscript, to orient the reader, we indicate associations where PGS are the independent variable with Beta_{PGS} , and where high-impact de novo variants are the independent variable with $\text{Beta}_{\text{denovo}}$

1.5. It appears that the sex-stratified analysis was conducted using PGS derived from GWAS not stratified by sex. This could likely introduce biases related to the sex distribution of the GWAS used to generate the PGS. The authors should stratify the iPSYCH-2015 by sex and generate female- and male-specific effects to generate their PGS. Additionally, more information should be provided regarding the age and sex characteristics of the iPSYCH-2015 cohort.

Thank you for this suggestion. We agree that sex-stratified analysis using sex-stratified autism GWAS may be more informative and easier to interpret. However, the current female-only autism GWAS is underpowered (4,845 autistic individuals, and 19,315 non-autistic individuals). Given this, we were aware that we may not be well-powered to identify modest effects. Nevertheless, we have now conducted sex-stratified over-transmission test using autism GWAS stratified by sex.

The results of the analyses are provided in Supplementary Table 17 but have been reproduced here below:

Males-only autism PGS				
		Meta-analysis		
Category	Sex	Mean (met	SE (meta)	N total
IQ > 70	Male	0.13	0.02	2919
IQ > 70	Female	0.22	0.04	430
Females-only autism PGS				
		Meta-analysis		
Category	Sex	Mean (met	SE (meta)	N total
IQ > 70	Male	0.07	0.02	2919
IQ > 70	Female	0.14	0.05	430

We have updated the results as follows:

“Additionally, in autistic individuals without co-occurring ID, females had ~ 100% overtransmission of autism PGS derived from the two sex-stratified GWAS (Supplementary Table 16). However, this overtransmission was statistically significant only for the PGS derived from the males-only GWAS.”

We have now provided additional information regarding age and sex of participants. In the Methods section, under polygenic scores, we have included the following lines:

“We restricted our polygenic score (PGS) associations to four GWAS. First, we included a GWAS of autism from the latest release from the iPSYCH cohort (iPSYCH-

2015)(Bybjerg-Grauholm et al.). This includes 19,870 autistic individuals (15,025 males and 4,845 females) and 39,078 individuals without an autism diagnosis (19,763 males and 19,315 females). All individuals included in this GWAS were born between May 1980 to December 2018 to mothers who were living in Denmark.”

1.6. The authors observed sex differences in the effect of high impact de novo variants, especially in those affecting DD genes. Did the authors verify whether this effect may be driven by X-linked loci?

Apologies that we did not make this clear in the manuscript. Our analyses for sex-differences in de novo variant burden was conducted using only autosomal genes. We have now made this clear in the results. Please see the updated text from the results below:

“The observed sex differences in high-impact *de novo* variants in autism may be explained entirely or partly by DD genes, which would suggest a biological mechanism shared by both autism and severe developmental disorders. We thus re-visited sex differences in high-impact *de novo* variants using data from SPARK and SSC (Supplementary Table 16), restricting our analyses to autosomal genes.”

1.7. The authors meta-analyzed the effects estimated across multiple cohorts. It would be important to report the heterogeneity estimates related to the meta-analyses conducted.

Thank you. We have now reported heterogeneity statistics (I^2 and Cochran's Q) in Supplementary Table 8. We have also updated our Results to report this. Please see this updated text in our response to Comment #1.2. Please note, we have only estimated heterogeneity statistics for the PGS analyses, and not for the de novo variant analyses as the latter was conducted only in two datasets, and hence underpowered for estimating heterogeneity using I^2 and Cochran's Q.

Reviewer #2:

Remarks to the Author:

With great interest I have reviewed the manuscript ‘Genetic correlates of phenotypic heterogeneity in autism’. In their manuscript, the authors provide valuable new insights that allow us to understand genetic heterogeneity in autism. The manuscript covers a broad scope of analyses of both common and rare genetic variation in relation to

autism. The analyses are sound and presented very well in the figures, and the manuscript is well-written. The analyses support their main message that a dimensional approach to autism aids in understanding its highly heterogeneous factors, which is a step forward from the heterogeneous case definitions that have been used in GWAS.

We thank the reviewer for these kind comments.

A review of the manuscript has left me with a couple of questions and remarks:

2.1. Results: Figure 1: panels B to D have 'mean score' on the y-axis. I find it difficult to interpret the effect size in these figures. I assume these are standardized effects? Would there be a way to put these mean score differences into perspective, i.e. give the reader a better idea of the magnitude of the observed effects?

Apologies. We did not standardise the scores before plotting the figures. We have done so now. Additionally, in Supplementary Table 4, we have now also provided Cohen's d as a measure of the effect size. We hope both the standardized scores and Cohen's d make it easier to interpret the effects.

We note that the sex differences are trivial in magnitude (Cohen's d ranging from 0.1 for Idiosyncratic repetitive speech and behaviour to 0.03 for Insistence of sameness factor). However, in contrast, Cohen's d was substantially higher when comparing the lowest IQ bin to the highest IQ bin (Cohen's d ranging from 0.67 for insistence on sameness to 0.96 for Sensory Motor behaviour). We did not calculate Cohen's d for age as it was a continuous metric.

2.2. Results: the authors report lower factor scores in older participants which may relate to recall bias of parent-reported scores in younger individuals. To what extent does selection bias play a role here? Since older participants often provide consent themselves (in contrast to children participants where parents consent), it may well be those adults with less severe symptoms that are willing/able to participate in a research study.

We agree that selection bias may play a role - it's unclear how and to what extent. All factor scores were derived from parent-report (and hence excluded adults). We do not think we can quite explore this in the current study due to the cross-sectional nature of the dataset. To this end, we have clarified this in the Results.

Amended Text in the Results:

“In this cross-sectional data, older participants had lower factor scores (i.e. fewer difficulties), with the exception of ‘Social Interaction’ (**Figure 1D**), in line with previous research³². Alternatively, this could reflect selection or diagnostic bias.”

2.3. Results: it would be of interest to know whether their dimensional approach actually changes gene finding efforts. This would be easy to do (but may be beyond the scope) by carrying out genome-wide association analyses and compare gene-mapping results/genetic correlations with the regular autism GWAS that is published.

Yes, we agree this would be interesting but is beyond the scope of this manuscript. It is unclear how we would interpret this in light of a few points. Second, GWAS results are influenced by both SNP heritability and sample sizes, both of which are low to conduct well-powered GWAS using the factor scores. Further samples may become available from SPARK which may enable such comparisons.

2.4. Results: PRS analyses: it is not quite clear from the text what discovery GWAS was used to obtain effect sizes for the PRS of autism (and schizophrenia, intelligence), please provide reference/sample size.

We have now provided references in the Results section. We have provided further details including the number of participants in the Methods section. We have also provided links to download the data in Supplementary Table 7.

2.5. Results: the lack of association between de novo variants and the autism scales is somewhat surprising. Might this relate to a selection effect as well? (i.e. lack of cases with highly penetrant/pathogenic mutations in the cohort). In any case, since these do not associate with autism dimensions it seems somewhat confusing to conclude that correcting PRS analyses for these non-significant effects means the PRS effects are independent.

This is a very valid point. We are not quite sure if this is due to selection. Indeed, Simon’s Simplex Collection should be enriched for individuals with highly penetrant genetic variants if there are any - this cohort was deeply phenotype and assessed for autism, and included only simplex families, so is enriched for *de novo* variants compared to the SPARK cohort. This is supported in our study and others (Antaki et al. 2021; Feliciano et al. 2019; Iossifov et al. 2014) - autistic individuals from the SSC are more enriched for de novo variants in highly-constrained genes compared to autistic individuals from SPARK. Another possibility is that because, by definition, we are sampling only at the extreme end of the autistic traits, we do not observe any significant

associations because we are truncating the underlying variance. Ultimately, addressing this point will require large samples from the general population with information on autistic traits.

However, we agree with your point regarding our conclusion. It is indeed meaningless to suggest that PGS and *de novo* affect autistic traits independently given that there is no effect of *de novo* variants. To address this, we have: (1) removed the line about independent effects between common and rare variants, and instead added a clause saying that lack of PGS effect size attenuation when including *de novo* variants was observed even for IQ (the only feature associated with common and rare variants); and (2) removed Figure 2C.

2.6. Results: Figure 2: the effect sizes of autism PRS with autism symptoms are rather small (assuming these are standardized effect sizes in a linear regression model). Could the authors comment on this, would this relate to the limited explained variance by PRS based on the latest GWAS?

Indeed. This is primarily due to the relatively low sample size of the autism GWAS compared to the other three GWAS, which will influence the variance explained by the autism PGS (currently, variance in autism explained by PGS of autism on the liability scale = 1.5%).

2.7. Results: Figure 3A: why did the authors choose to bin IQ on the y-axis? (in unequal bins)

This was because full-scale IQ measured in SPARK was in bins. As SPARK was the largest dataset used in the study, consequently, we binned full-scale IQs from SSC as well to match what was available in SPARK to create Figure 3A. In the SSC, we obtain similar figures when using either continuous IQ scores or binned IQ scores, suggesting that the results do not differ due to binning.

2.8. Results: the observation of a lower PRS in carrier autistic individuals compared to non-carriers is an interesting finding. How much of this may relate to collider bias between the two?

Thank you for raising this point. This is true. In the general population, one might expect the autism PGS and presence of high-impact *de novos* to be independent. However, because we have conditioned on autistic individuals, we may have induced a negative correlation between them because people have to have either a high PGS or large-effect *de novo* to cross the threshold.

Having said that, one might actually expect there to be a weak positive correlation in the general population between the autism PRS and number of *de novos*. This is because men with more autistic traits might tend to have children later, so their children may have both high autism PGS and an increased likelihood for *de novo* variants.

In the current study we are unable to delineate between the two hypotheses. However, we have acknowledged this in the Discussion:

“However, this negative correlation between high-impact *de novo variants* and autism PGS may not extend to the general population. Because we have conditioned on autistic individuals, we may have induced a negative correlation between them because people have to have either a high PGS or high-impact *de novo variants* to cross the diagnostic threshold.”

Decision Letter, first revision:

Our ref: NG-A58080R1

9th December 2021

Dear Varun,

Your revised manuscript "Genetic correlates of phenotypic heterogeneity in autism" (NG-A58080R1) has been seen by the original referees. As you will see from their comments below, they find that the paper has improved in revision, and therefore we will be happy in principle to publish it in Nature Genetics as an Article pending final revisions to comply with our editorial and formatting guidelines.

We are now performing detailed checks on your paper and we will send you a checklist detailing our editorial and formatting requirements soon. Please do not upload the final materials or make any revisions until you receive this additional information from us.

Thank you again for your interest in Nature Genetics. Please do not hesitate to contact me if you have any questions.

Sincerely,
Kyle

Kyle Vogan, PhD
Senior Editor
Nature Genetics
<https://orcid.org/0000-0001-9565-9665>

Reviewer #1 (Remarks to the Author):

The authors adequately addressed my previous concerns.

Reviewer #2 (Remarks to the Author):

The authors have addressed my concerns in the current version of the manuscript, I have no further comments.

Final Decision Letter:

In reply please quote: NG-A58080R2 Warriier

1st April 2022

Dear Varun,

I am delighted to say that your manuscript "Genetic correlates of phenotypic heterogeneity in autism" has been accepted for publication in an upcoming issue of Nature Genetics.

Your paper will be published online after we receive your corrections and will appear in print in the next available issue. You can find out your date of online publication by contacting the Nature Press Office (press@nature.com) after sending your e-proof corrections. Now is the time to inform your Public Relations or Press Office about your paper, as they might be interested in promoting its publication. This will allow them time to prepare an accurate and satisfactory press release. Include your manuscript tracking number (NG-A58080R2) and the name of the journal, which they will need when they contact our Press Office.

Before your paper is published online, we will be distributing a press release to news organizations worldwide, which may very well include details of your work. We are happy for your institution or funding agency to prepare its own press release, but it must mention the embargo date and Nature Genetics. Our Press Office may contact you closer to the time of publication, but if you or your Press Office have any enquiries in the meantime, please contact press@nature.com.

Please note that Nature Genetics is a Transformative Journal (TJ). Authors may publish their research with us through the traditional subscription access route or make their paper immediately open access through payment of an article-processing charge (APC). Authors will not be required to make a final decision about access to their article until it has been accepted. [Find out more about Transformative Journals](https://www.springernature.com/gp/open-research/transformative-journals)

Authors may need to take specific actions to achieve [compliance](https://www.springernature.com/gp/open-research/funding/policy-compliance-faqs) with funder and institutional open access mandates. If your research is supported by a funder that requires immediate open access (e.g. according to [Plan S principles](https://www.springernature.com/gp/open-research/plan-s-compliance)), then you should select the gold OA route, and we will direct you to the compliant route where possible. For authors selecting the subscription publication route, the journal's standard licensing terms will need to be accepted, including [self-archiving-and-license-to-publish](https://www.nature.com/nature-portfolio/editorial-policies/self-archiving-and-license-to-publish). Those licensing terms will supersede any other terms that the author or any third party may assert apply to any version of the manuscript.

Please note that Nature Research offers an immediate open access option only for papers that were first submitted after 1 January 2021.

You can now use a single sign-on for all your accounts, view the status of all your manuscript

submissions and reviews, access usage statistics for your published articles and download a record of your refereeing activity for the Nature journals.

If you have not already done so, we invite you to upload the step-by-step protocols used in this manuscript to the Protocols Exchange, part of our on-line web resource, natureprotocols.com. If you complete the upload by the time you receive your manuscript proofs, we can insert links in your article that lead directly to the protocol details. Your protocol will be made freely available upon publication of your paper. By participating in natureprotocols.com, you are enabling researchers to more readily reproduce or adapt the methodology you use. [Natureprotocols.com](https://natureprotocols.com) is fully searchable, providing your protocols and paper with increased utility and visibility. Please submit your protocol to <https://protocolexchange.researchsquare.com/>. After entering your [nature.com](https://www.nature.com) username and password you will need to enter your manuscript number (NG-A58080R2). Further information can be found at <https://www.nature.com/nature-portfolio/editorial-policies/reporting-standards#protocols>

Sincerely,
Kyle

Kyle Vogan, PhD
Senior Editor
Nature Genetics
<https://orcid.org/0000-0001-9565-9665>